# Composition and metabolism of microbial communities in soil pores

Zheng Li [1], Alexandra N. Kravchenko [2,3] ✉, Alison Cupples[1], Andrey K. Guber[2,3], Yakov Kuzyakov[4], G. Philip Robertson [3,5] & Evgenia Blagodatskaya[6]

Delineation of microbial habitats within the soil matrix and characterization of their environments and metabolic processes are crucial to understand soil functioning, yet their experimental identification remains persistently limited. We combined single- and triple-energy X-ray computed microtomography with pore specific allocation of $^{13}C$ labeled glucose and subsequent stable isotope probing to demonstrate how long-term disparities in vegetation history modify spatial distribution patterns of soil pore and particulate organic matter drivers of microbial habitats, and to probe bacterial communities populating such habitats. Here we show striking differences between large (30-150 μm Ø) and small (4-10 μm Ø) soil pores in (i) microbial diversity, composition, and life-strategies, (ii) responses to added substrate, (iii) metabolic pathways, and (iv) the processing and fate of labile C. We propose a microbial habitat classification concept based on biogeochemical mechanisms and localization of soil processes and also suggests interventions to mitigate the environmental consequences of agricultural management.

Soil is a crucial component of all terrestrial ecosystems, the main source of human food production, and one of the major mediators of atmospheric $CO_2$ level[1,2]. Soil functioning is enabled by a complex community of microorganisms[3] inhabiting an intricate physical frame of soil pores[4,5], adapting their life and C-acquisition strategies[6] to highly variable micro-environmental conditions[7,8].

Plant roots are among the main drivers of soil micro-environments[9,10], and vegetation community composition and diversity substantially affect formation of microhabitats within the soil matrix[11]. Plant roots form and modify soil pore structure, i.e., pore-size distributions, pore shapes and surface properties, and pore connectivity[12], while pore structure in-turn impacts soil microorganisms[13–15]. Greater plant species richness increases root biomass and diversifies root residue inputs[16], providing more organic C for soil microorganisms[17] and raising microbial activity[17–19]. Plant communities, especially the diverse perennials, strongly influence soil

pores−form biopores, leading to structural changes positively associated with soil organic matter accumulation[20].

The pore structure and distribution as well as fluxes of water within pores ultimately define oxygen and nutrient supply for microbial functioning[21,22]. Pores in the <10 μm Ø size range are often saturated by water, which limits oxygen supply, while pores in the >1000 μm Ø are mostly unsaturated and even dry[23]. Pores with diameters in the few-tens to couple-hundred-micron size range provide an optimal balance of oxygen, water, carbon (C), and nutrient inflows for resident microorganisms. Therefore, microbial communities inhabiting pores of various sizes differ in their composition, life strategies, and activities: >30 μm Ø pores can better stimulate fast decomposition of newly added C and have a greater abundance of certain microbial taxa, as compared to <10 μm Ø pores[24–26]. Yet, even this is a simplification, subject to actual proximity to substrates, connectivity, and the presence and accessibility of nearby water[27]. Water as

[1]Department to Civil and Environmental Engineering, Michigan State University, East Lansing, MI, USA. [2]Department of Plant, Soil and Microbial Sciences, Michigan State University, East Lansing, MI, USA. [3]DOE Great Lakes Bioenergy Research Center, Michigan State University, East Lansing, MI, USA. [4]Department of Soil Science of Temperate Ecosystems, Department of Agricultural Soil Science, University of Göttingen, Göttingen, Germany. [5]W. K. Kellogg Biological Station, Michigan State University, Hickory Corners, MI, USA. [6]Helmholtz Centre for Environmental Research, Halle, Germany. ✉e-mail: kravche1@msu.edu

films and menisci between soil particles provide habitable niches and crevasses for microorganisms to grow, function, move, and interact[28–31].

Thus the movement of soil solution, and, in particular, pore-scale hydraulic connectivity, is a key element influencing bacteria and especially their access to organic matter and subsequent metabolism[32–34], with ultimate consequences for soil carbon accretion and stability and for ecosystem processes such as decomposition and denitrification[22,35]. Yet we know very little about the impact of hydraulic connectivity on microbes at the scale of individual soil pores. While the expansion of X-ray computed microtomography (μCT) has made it possible to assess and characterize pores in intact soil cores[36,37], the visualization of water has been much more challenging and therefore, not yet combined with pore-size visualization for assessing the full impact of pore structure on ecosystem processes.

Here we document the combined effects of long-term differences in vegetation history on the spatial distribution of soil pores and particulate organic matter (POM) as well as on the hydraulic connectivity of small (4–10 μm Ø) and large (30–150 μm Ø) pores within intact soil matrices. We further test the degree to which vegetation history affects bacterial richness, community composition, and metabolism, specifically in small (4–10 μm Ø) vs. large (30–150 μm Ø) pores within the soil matrix, with consequent effects on soil C processing. We used three vegetation systems of contrasting managements and plant diversities established in a replicated blocked field experiment: (i) a multiyear fallow followed by 2 years of monoculture corn (*Zea mays* L.), which for brevity we will refer to as bare soil, (ii) a perennial monoculture switchgrass (*Panicum virgatum* L.) community, and (iii) a polyculture restored prairie community of >20 native North American grasses and forbs. We used μCT to characterize soil pore structure and localize and quantify the spatial distribution of particular organic matter (POM), employing a triple-energy μCT approach to examine pore-level spatial patterns of water distributions. We simulated labile substrate additions by applying small quantities of labeled glucose—an abundant component of root exudates and decomposition product of carbohydrates that is metabolized by the majority of soil microorganisms with well-defined uptake mechanisms and mineralization pathways[38]. We then used stable isotope probing ($^{13}$C-DNA/RNA-SIP)[39–41] to identify microorganisms actively assimilating the glucose-derived C ($^{13}$C) into nucleic acids[42–44]. Results demonstrate that microbial habitats defined by soil pores and POM of various plant communities differ in ways that strongly influence microbial composition and activity, and thus may impact ecosystem processes such as decomposition, nitrogen processing, and carbon sequestration.

## Results

### Soil chemical and physical characteristics

The soil under restored prairie developed higher C and N contents, lower bulk density, and larger total porosity than the switchgrass soil (Supplementary Table S2). The average distance to pores tended to be the lowest in the prairie soil (Fig. 1c and Supplementary Table S2), suggesting that the distribution of pores through the matrix of the prairie soil was the most uniform of the three systems.

Large (30–150 μm Ø) and small (4–10 μm Ø) pores in soils of all vegetation systems markedly differed in their connectivity. The large pores not connected to the main pore space constituted 0.1% of the total soil volume (Fig. 1d), i.e., amounted to just a small fraction of all large pores (3.8–5.3%, Supplementary Table S2). In contrast, out of the total 1.5–1.9% of the soil volume occupied by small pores (Supplementary Table S2), >60% were disconnected from the main connected pore space (Fig. 1d).

The two perennial vegetation systems, i.e., switchgrass and prairie, contained approximately five times more POM than the bare soil system (Supplementary Table S2). The average distance to POM fragments in the prairie soil was only half of that in the switchgrass

(Fig. 1c and Supplementary Table S2), indicating a much more homogeneous distribution of POM within the prairie soil's matrix. The size of POM fragments in the prairie soil was, on average, smaller by one-third as compared to switchgrass (Fig. 1e), and the prairie soil had >5 times more POM fragments than the switchgrass soil (Fig. 1f). The high content of uniformly distributed POM fragments resulted in most of the total volume of the prairie soil (i.e., 87% of it) to be in close proximity (<300 μm) to and, thus, under the direct influence of POM. Only 48% and 30% of the soil volumes of the switchgrass and bare soils, respectively, were within <300 μm distance to POM (Fig. 1g).

### Dopants and glucose additions to small and large pores

Analyses of the multi-energy images from μCT scans revealed that the dopant solutions reached the target pores (Fig. 1b). The small (4–10 μm Ø) pores were filled by the added KI solution to ~58% saturation; the saturation of the large (30–150 μm Ø) pores by $BaCl_2$ was 10–15% (Supplementary Fig. S2). Given the high variability and connectedness of the soil pore space, we were not surprised that some portions of the solutions were found in the pores of intermediate size (10–30 μm Ø), and there was some small overlap in saturation between the small and large pores. However, the liquid intended for the large pores did primarily occupy pores >50 μm Ø, with a peak saturation around 100–150 μm, which was within the target size.

The solution added to the large pores formed sizeable menisci between soil particles as well as thick water films on the boundaries of very large pores, which had their central portions filled with air (Fig. 1b). The space filled with the liquid intended for the large pores was much better connected than that filled with the solution added to the small pores. Specifically, the largest connected pore volume filled with large-pore-targeting solution occupied 2.5% of the total μCT-visible pore space (>4 μm Ø pores). The largest connected pore volume filled with small-pore-targeting solution occupied only 0.4% of the pore space.

### Glucose originated carbon

The switchgrass and prairie systems contrasted in the temporal dynamics of $^{13}$C-$CO_2$: on the 1st day of the incubation, the $^{13}$C enrichment was higher in the $CO_2$ released from the prairie soil, while during the subsequent few days (days 3 and 7) the $^{13}$C enrichment was higher in the switchgrass (Fig. 2a and Supplementary Table S3). On the first day after glucose input, $^{13}$C atom-% of the $CO_2$ released from the large pores of switchgrass soil tended to be lower than that from small pores.

After the first 24 h of glucose utilization, the total $^{13}$C amounts remaining in the large and small pores were similar (Fig. 2b). Approximately 45% of the total $^{13}$C recovered within the switchgrass soil after 24 h was present as DOC (Fig. 2c), while within the prairie soil it was only ~18%. Using a conversion factor of 0.45 for the fumigation approach to microbial biomass determination, the $^{13}$C incorporation into microbial biomass constituted on average ~33% of the $^{13}$C remaining in the soil (Fig. 2d).

After the 30-day incubation, the $^{13}$C remaining in the soil was higher in the small compared to the large pores (Supplementary Table S3), with the difference especially pronounced in the switchgrass soil. The dissolved organic $^{13}$C constituted only ~2% of the total $^{13}$C recovered in the prairie soil, and was even lower (<1%) in the small pores of the switchgrass cores. $^{13}$C of microbial biomass constituted ~20% and 12% of the total soil $^{13}$C in the prairie and switchgrass soils, respectively.

### Microbial community analysis

The perennial vegetation was a major determinant of the microbial community composition (Supplementary Figs. S3 and S4). Compared to the bare soil, the relative abundance of phyla Latescibacteria, Gemmatimonadetes, *Planctomycetes*, Proteobacteria, and

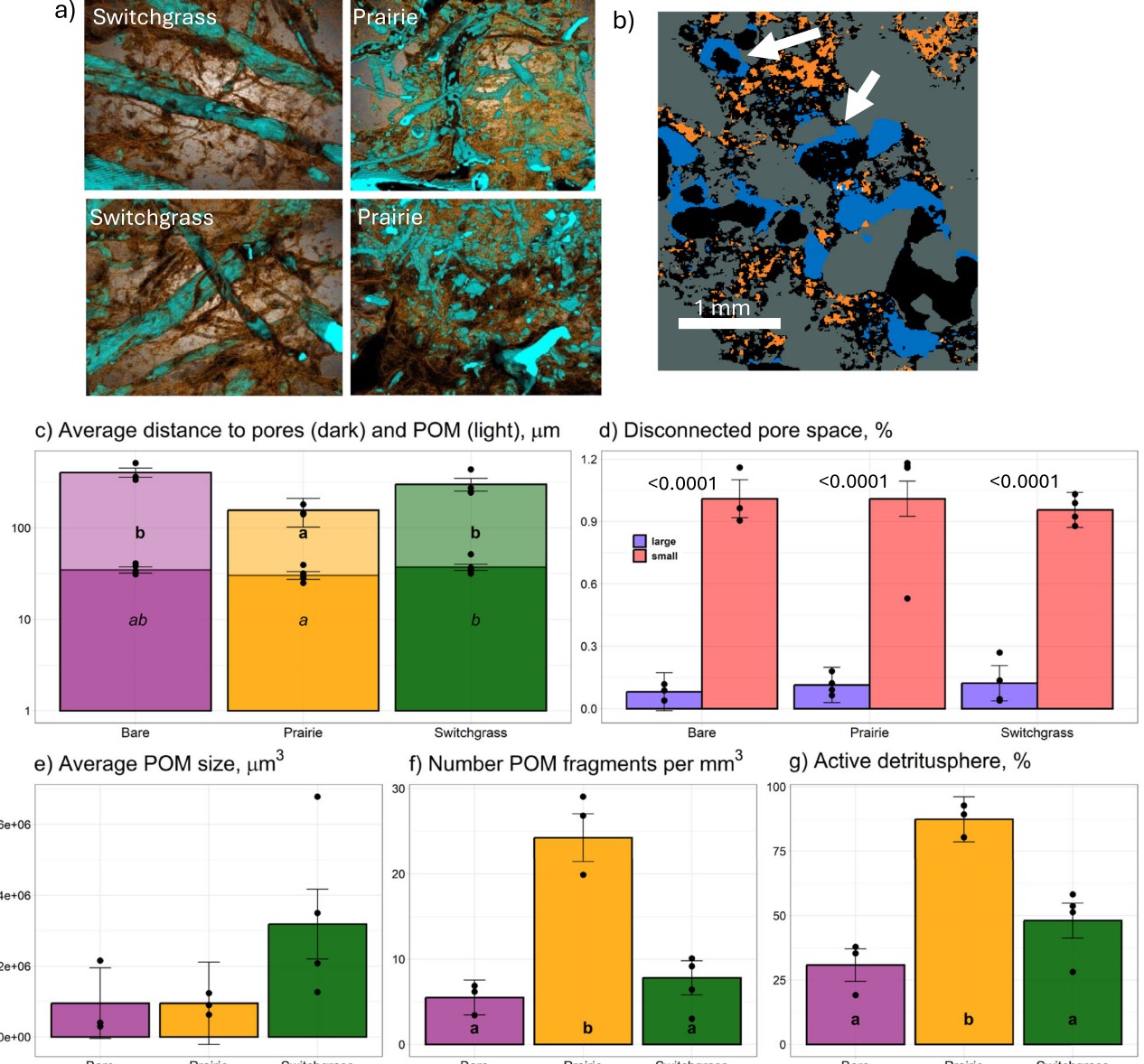

**Fig. 1 | Images and characteristics of soil pores >4 μm Ø and particulate organic matter (POM) within the soils of three experimental plant communities.** **a** Sample images with pores (brown) and POM (cyan) from selected representative soil cores of prairie and switchgrass soils. **b** Visualization of the liquid added to small (4–10 μm Ø) (orange) and large (30–150 μm Ø) (blue) pores. White arrows are pointing to examples of menisci between soil particles and water films within the large pores. **c** Average distances to pores (dark shading) and POM (light shading) in the soils of the three communities. **d** Volumes of large and small pores that were not connected to the largest connected pore cluster (% of total soil volume). **e** Average size of POM fragments. **f** Number of POM fragments per unit of soil volume. **g** The percent of the total solid volume that was within 300 μm of POM, and thus considered to be active detritusphere. For all panels, shown are means, standard errors as error bars, and original data as dots. The letters on (**c**, **f**, **g**) mark the differences among the vegetation systems significant at $P < 0.05$ (bold) or $P < 0.1$ (italic). The results of the two-sided tests for the differences between the large and small pores are reported with $P$ values shown above bars on (**d**). Source data for (**c–g**) are provided as a Source Data file.

*Verrucomicrobia* (Supplementary Fig. S4) was greater in the prairie and switchgrass systems. In contrast, the relative abundances of oligotrophic (*Acidobacteriales*), slow-growing (*Frankiales*), pseudomycelium-forming (Actinomycetota), spore-forming (*Ktedonobacterales*), and desiccation-resistant (Firmicutes) bacteria were higher in bare versus planted soil. Lower pH of the bare soil might have been a contributor to a greater abundance of *Acidothermus*.

All diversity indices pointed to a lower microbial community diversity in the bare soil as compared to the prairie and switchgrass soils (Fig. 3a and Supplementary Table S4). Across both incubation times, the microbial diversity was higher in the prairie than in switchgrass soil, and at the end of the incubation, it was higher in the small

than in the large pores (Fig. 3a, $P < 0.05$). Microbial biomass was higher in the prairie soil than in the soils of bare and switchgrass systems (Supplementary Table S2).

### $^{13}$C-enriched phylotypes: immediate response to glucose addition (24-h incubation)

The total number of OTUs that responded to $^{13}$C-glucose within 24 h was substantially greater in the switchgrass and bare soil as compared with the prairie soil (64 OTUs) (Fig. 3b). *Pseudomonas* was the group with the highest $^{13}$C enrichment in the large pores of all three systems, with the $^{13}$C enrichment in the prairie soil twice as high as that in any other group (Fig. 3c and Supplementary Table S5). There were several

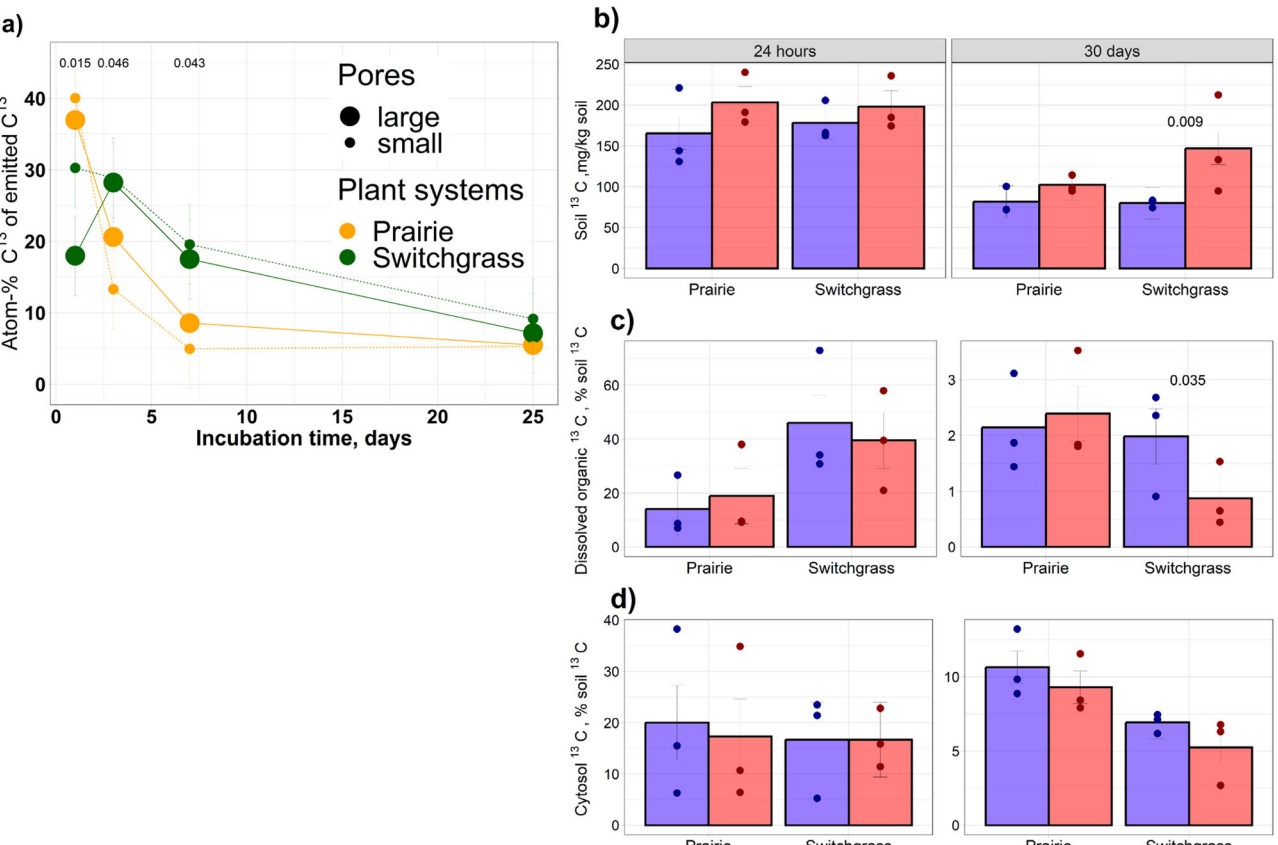

**Fig. 2 | Fate of $^{13}$C-glucose carbon added to small (4–10 μm Ø) and large (30–150 μm Ø) pores of the prairie and monoculture switchgrass communities.** **a** Atom-% $^{13}$C of emitted $CO_2$ during the 30-day incubation of the intact samples. Shown are means and standard errors as error bars. *P* values are shown for the differences between the two systems significant at $P < 0.05$. Total (**b**), dissolved organic (**c**), and cytosol (**d**) C of glucose-origin ($^{13}$C) remaining in the soil after 24 h and 30-day incubation. Shown are means, standard errors as error bars, and original data as dots. Significant ($P < 0.05$) differences between large and small-pore treatments within the system and incubation times as determined via simple-effect *F* tests are marked, with *P* values shown above the bars. Source data are provided as a Source Data file.

groups of highly $^{13}$C-enriched bacteria that were plant system-specific (Fig. 3c and Supplementary Table S5). The taxa *Planococcaceae* and *Micrococcaceae* incorporated $^{13}$C only in the bare soil, while *Cupriavidus* and *Duganella* incorporated $^{13}$C only in the switchgrass soil. *Clostridium* took the $^{13}$C in the large and, especially, small pores of prairie and switchgrass.

### $^{13}$C-enriched phylotypes: short-term response to glucose addition (30-day incubation)

In contrast to the immediate response to the glucose input, the total number of OTUs enriched with $^{13}$C after a 30-day incubation was almost three times larger in the prairie (259 OTUs) than in the switchgrass soil (90 OTUs) (Fig. 3b). While only few taxa remained $^{13}$C-enriched or even increased their $^{13}$C enrichment with time, several new groups became enriched in all pores and vegetation systems (Fig. 3c and Supplementary Table S6). *Cupriavidus* and *Duganella* remained enriched in the large and small pores of the switchgrass soil, while absent in the prairie. *Bradyrhizobium* and *Bdellovibrio* were among the top-enriched bacteria found only in the prairie but not in the switchgrass soil (Supplementary Table S6).

Approximately 75% of the $^{13}$C-enriched bacteria were present in both large and small pores in the prairie soil, while only 48% were present in both large and small pores in the switchgrass soil (Fig. 3b). *Cellvibrio* and *Bacteriovorax* were among the organisms remaining $^{13}$C-enriched in the large pores (Supplementary Table S7). *Pseudomonas* remained the most (prairie) or the second most (switchgrass) enriched bacteria in the large pores, yet its $^{13}$C enrichment was much

lower in the small pores of both systems (Supplementary Table S6). Several members of *Bdellovibrio* group were enriched in the large pores of the prairie soil, while this group had only very low $^{13}$C incorporation in the small pores (Supplementary Table S6).

### $^{13}$C-enriched functional genes: glycolysis/gluconeogenesis and citric acid cycle

The lack of $^{13}$C enrichment in genes responsible for the first step of glycolysis in the large pores of the switchgrass system ([EC:2.7.1.2], [EC:2.7.1.63]) was noted in both 24 h and 30-day incubations, while the relevant genes were enriched in the small pores (Supplementary Fig. S5). On the contrary, the $^{13}$C enrichment was common in large and small pores of the prairie soil. In the 24 h incubation, the genes encoding pyruvate dehydrogenase ([EC:1.2.4.1], [EC:2.3.1.12]), the enzyme responsible for conversion of pyruvate to acetyl-CoA, were $^{13}$C-enriched in the switchgrass but not in the other two systems.

In the prairie soil, almost all the genes involved in glucose-to-pyruvate conversion steps of glycolysis were enriched in both small and large pores at both incubation times. In the switchgrass system, the genes responsible for glucokinase production were enriched only in the small-pore treatment in both the 24 h and 30-day incubation. The genes encoding the enzymes involved in dihydrolipoamide dehydrogenase [EC:1.8.1.4]) were activated in the small pores of bare soil in the 24-h incubation (Supplementary Figs. S5a and S6a). In the 30-day incubation, the genes encoding the enzymes involved in pyruvate metabolism (pyruvate dehydrogenase, dihydrolipoamide dehydrogenase [EC:1.8.1.4]) were enriched only in the small pores of both

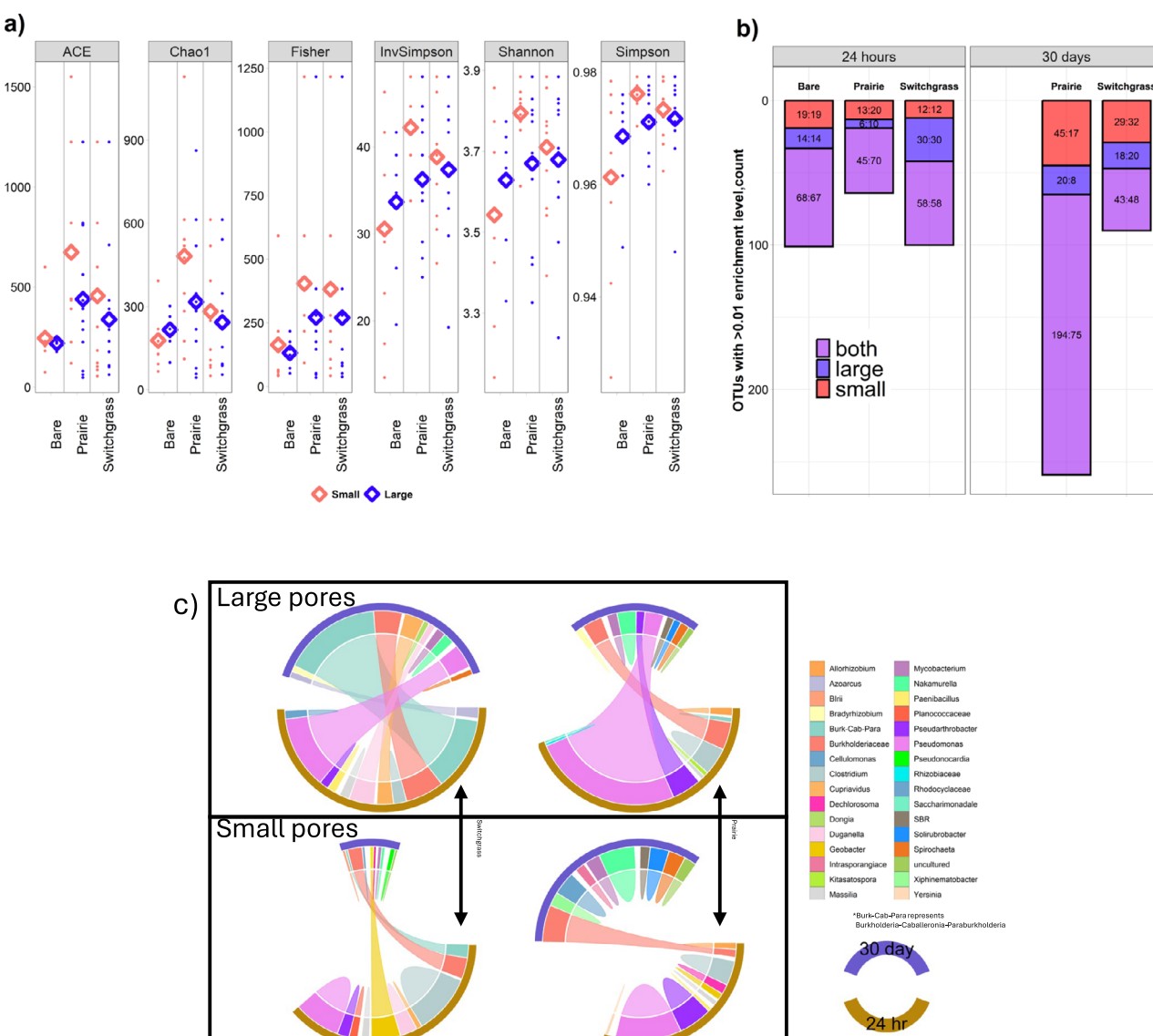

**Fig. 3 | Microbial community characteristics after 24 h and 30 d of incubation in the soils of the three plant communities, with glucose placed either in large or in small pores. a** Alpha diversity assessed by richness (Chao1, ACE) and diversity (Shannon, Simpson, Inverse of Simpson and Fisher) indexes. Note: the hollow diamonds represent the means of the indices in control, large pore, and small-pore incubation treatments, averaged across both incubation times for prairie and switchgrass. **b** Numbers of operational taxonomic units (OTUs) that responded to glucose addition that were found either only in large, small, or in both pore-size treatments. Shown on each bar are OTU counts (first number) and percent of the total number of OTUs (second number). **c** Top ten genera that were [13]C-enriched in the prairie and switchgrass plant systems in 24 h and 30-day incubations. The width of the base for each genus is proportional to its relative enrichment level. Source data for (**a**, **b**) are provided as a Source Data file.

switchgrass and prairie systems (Supplementary Fig. S6b). While in the 24-h incubation, the genes responsible for pyruvate fermentation to ethanol via alcohol dehydrogenase ([EC:1.1.1.1], [EC:1.1.2.8]), an anaerobic metabolism, were enriched both in the large and small pores of the switchgrass, after the 30-day incubation they were only enriched in the switchgrass's small pores.

### [13]C-enriched functional genes: nitrogen metabolism
As indicated by the [13]C enrichment of the nirK gene, the bacteria benefiting from added glucose during the first 24 h of incubation performed denitrification in the switchgrass, but not in any other systems (Supplementary Fig. S7). While later (after 30 days), several genes involved in coding denitrification enzymes, namely, napA, nirK, and nosZ, were enriched in the prairie, but not in the switchgrass system.

Genes encoding carbonic anhydrase ([EC:4.2.1.1]), the enzyme that converts $CO_2$ and water into carbonic acid, and nitrite reductases ([EC:1.7.1.15], [EC:1.7.2.2]), the enzymes involved in dissimilatory nitrate reduction, were enriched in small pores of both prairie and switchgrass. Small pores of the switchgrass system had enrichment of the gene encoding nitrate/nitrite transport complex (nrtA) in the 24 h of incubation (Supplementary Fig. S7a) and also of the gene involved in N fixation nitrogenase (nifG) at both incubation times (Supplementary Figs. S7a and S7b). None of these genes were active in any other vegetation systems.

In the 24-h incubation, in both prairie and switchgrass and in the 30-day incubation in the prairie, the $CO_2$-fixing gene [EC:4.2.1.1] was [13]C-enriched only in the large pores (Supplementary Figs. S7a and S7b).

In the 30-day incubation, glutamate dehydrogenase genes ([EC:1.4.1.2], [EC:1.4.1.4]) were enriched only in the large pores of both

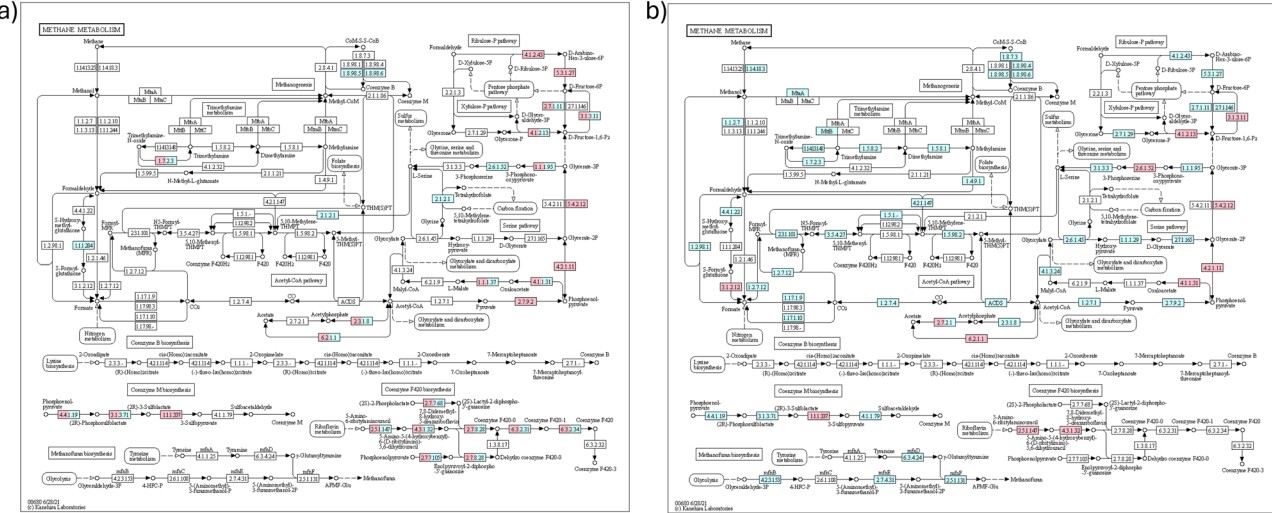

**Fig. 4 | Methane metabolism genes enriched in in the soil of the studied systems after 30-day incubations.** Shown are the genes enriched in the prairie (**a**) and switchgrass (**b**) communities. Blue and pink denote enrichments in small and large pores, respectively. Diagrams created with KEGG (Copyright Permission 230979).

systems. Yet, glutamate synthesis ([EC:1.4.1.13], [EC:1.4.7.1]) was taking place in large and small pores of both systems.

### $^{13}$C-enriched functional genes: methane metabolism

After 24 h and 30 days, there were many enriched genes associated with the methane metabolism in the switchgrass system as compared to the prairie (Figs. 4 and S8). In switchgrass soil, gene encoding anaerobic carbon-monoxide dehydrogenase (E.1.2.7.4) was enriched in the large pores at 24-h incubation and in the small pores after 30 days. The glucose localization effects differed between the three soil–plant systems and overall, methane metabolism mainly occurred in the switchgrass and in the small pores (Fig. 4).

## Discussion

Multiple years of dissimilar vegetation not only affected soil C and N contents but led to contrasting micro-scale patterns in the spatial distribution of soil pores and organic residues. As expected, many bacterial generalist taxa commonly found in soils worldwide[45,46] were also detected in the soils of the studied systems. Yet the microbial groups residing in large vs. small soil pores responded differently to the resource addition. The responses differed both in terms of life strategies and in the metabolic pathways they employed, with implications for the fate and protection of the added glucose, and, as inferred from genetic analysis, also for other metabolic processes such as denitrification and methanogenesis.

Differences in POM spatial distributions (Fig. 1c, e, f) might be the key driver of the reported differences in decomposition between soils of monoculture switchgrass and prairie[47]. A ubiquitous spread of small POM fragments in the prairie soil positioned most of the prairie soil's matrix within an active detritusphere, i.e., <300 μm distance from POM. Since many of such POM fragments were former roots, the 1–2 mm areas around them were also previously within an active rhizosphere. The detritusphere experienced major C and nutrient influxes from the decomposing organic residues[48,49] and, in the prairie soils, acted as one giant contiguous hotspot of microbial activity[50]. The large sizes of POM pieces in switchgrass (-3.5 × 10$^6$ μm$^3$) as opposed to prairie (-0.9 × 10$^6$ μm$^3$) (Fig. 1a, e), and large average distances to POM in switchgrass (Fig. 1c), suggest that a substantial portion of the switchgrass soil matrix was not under the direct influence of either detritusphere or former rhizosphere, thus, was likely devoid of fresh nutrient inputs. Likewise, most of the matrix in the bare soil was neither in the current detritusphere nor former rhizosphere. This result is especially notable because POM comparisons among plant

communities commonly focus only on total POM contents rather than on spatial distribution patterns[51]. Our findings (Supplementary Table S2) demonstrate that even though soils from switchgrass and restored prairie systems can have similar POM contents (e.g., ref. 52), their POM spatial distribution patterns can differ substantially, generating diverging impacts on soil C processing and microbial communities.

Microbial response to new substrate and implications for C processing differed in the immediate (i.e., 24 h) and longer (30 d) duration. Initially, i.e., 24 h after the application, the glucose addition approach equalized conditions within the micro-environments to which the glucose was added. The same quantities and concentrations of the glucose solutions, which were made using dH$_2$O with high oxygen levels, were added to both large and small pores. Moreover, the oxygen/glucose-rich solution was pushed into the small pores, which does not happen under natural conditions. We surmise that the resultant burst of microbial activity in both large and small pores probably led to similarly sizeable collapses in O$_2$ concentrations. This explanation is supported by (1) the abundance of anaerobic organisms among the initially enriched taxa, e.g., *Clostridium* (Supplementary Table S5), enrichment in several genes responsible for anaerobic processes, including the anaerobic pyruvate to ethanol pathway in both large and small pores (Supplementary Fig. S5a), and anaerobic carbon-monoxide dehydrogenase activity in the large pores of the switchgrass (Supplementary Fig. S8). Not surprisingly, the same newly added C source and the same resultant environmental conditions within 24 h of glucose addition led to the same groups, i.e., *Pseudomonas*, *Burkholderiaceae*, *Pseudarthrobacter*, and *Clostridium*, being among the top glucose consumers, heavily enriched in both large and small pores of bare, prairie, and switchgrass soils (Supplementary Table S5).

Despite these similarities, inherent differences in microbial community size and composition of soils under the three vegetation systems led to differences in processing new C. Over the years of bare soil management and monoculture switchgrass cultivation, the microbial communities apparently became less diverse as they adapted either to overall limitations in new inputs in bare soil or to a narrow range of C sources of switchgrass roots and rhizodeposition in the switchgrass system[53]. In contrast, the high plant diversity of the prairie system led to a greater diversity in chemical composition and types of root inputs[54] and to a greater heterogeneity of the pore space, together maximizing microbial diversity in its small pores (Fig. 3a). Thus, as can be inferred from low $^{13}$CO$_2$ emissions and high dissolved organic $^{13}$C

concentrations in the switchgrass soil at the start of the incubation (Fig. 2a), the microbes there were delayed in utilizing the new resource. In contrast, the diverse and large microbial community of the prairie soil immediately reacted to glucose addition. The glucokinase activity in the prairie soil might have given a competitive advantage for some bacterial groups in glucose consumption (Supplementary Figs. S5 and S6). Lack of such activity from the inhabitants of the large pores of the switchgrass soil is probably what resulted in a slow utilization of the newly added glucose.

After 30 days of incubation the micro-environmental conditions in the soil matrix likely recovered to their original natural state. That is, after the initial oxygen depletion in the large pores of both switchgrass and prairie systems, the exchange with the atmosphere would have restored oxygen concentrations, allowing aerobic microbial groups to flourish. The oxygen-tolerant initial top consumers of $^{13}$C-glucose remained highly $^{13}$C-enriched until day 30, which reflected their slow turnover and low cross-feeding intensity in the community (Fig. 3c and Supplementary Tables S5 and S6).

In the small pores of the switchgrass soil, the micro-environmental conditions changed substantially from immediately after the glucose addition, bringing marked shifts in the community composition of $^{13}$C-enriched organisms. We surmise that because of the poor contact with the atmosphere the initial oxygen depletion due to active glucose oxidation was not relieved in the small pores. While the glucose resource disappeared the anoxic conditions still proliferated, prompting processes such as anaerobic fermentation of pyruvate to ethanol (Supplementary Fig. S5b), dissimilatory nitrate reduction (Supplementary Fig. S7b), and methanogenesis (Fig. 4). The remaining necromass apparently was poorly utilized by the organisms that succeeded the first glucose responders (Fig. 3c). The number of these secondary $^{13}$C consumers decreased and their $^{13}$C enrichment dropped as compared to those at 24 h after glucose input (Fig. 3c). It seems probable that the microbial community of the small pores in switchgrass soil shifted to dormancy or basic maintenance after glucose was exhausted, as opposed to active successional development and cross-feeding.

In the prairie soil, more than 100 previously non-enriched bacterial groups now acquired the $^{13}$C label in both large and small pores (Fig. 3b). The extracellular metabolites and necromass of the first responding community were heavily utilized by a very large and diverse group of successive organisms (Fig. 3b, c and Supplementary Table S6). More of the added label was still embedded in living microbial biomass or was in a form of DOC in the soil solution, while less of it was part of the $^{13}$C remaining in soil as compared with small pores of switchgrass (Fig. 2b–d).

A possible explanation for the similarity between the large and small pores in the prairie soil is that most of the soil matrix harboring small pores belonged to the detritusphere (Fig. 1f and Supplementary Table S2). Thus, regardless of whether glucose was added to large or small pores in the prairie soil, it was consumed by the detritusphere microorganisms, well adapted to benefit from the labile resources both immediately (Fig. 2d) and in the medium-term. In contrast to the switchgrass system, the community of secondary decomposers in small pores of prairie remained as highly enriched after the 30-day incubation as they were 24 h after glucose input (Fig. 3c). Intensive microbial turnover resulted in strong successional changes in the community, yet the extra resource (i.e., C from the added glucose) was re-utilized by ever larger groups of organisms.

Several processes involved in N metabolism tended to be system- and pore-specific (Supplementary Fig. S7), suggesting differences in N availability and the adaptations of resident microbial communities. We surmise that N availability was low in the small pores, especially in the switchgrass soil, prompting a greater diversity of routes of N procurement by their bacterial inhabitants. For example, small pores of the switchgrass soil were populated by (i) bacteria with fast N uptake,

as suggested by an enrichment of the gene coding for nitrate/nitrite transport complex (nrtA) in the 24-h incubation, and (ii) bacteria involved in N fixation, as suggested by the gene involved in N fixation nitrogenase (nifG) in both incubation times. On the other hand, the glutamate dehydrogenase pathway, which is utilized for glutamate synthesis under conditions of N excess[55], was active in the large, but not small, pores of both systems. The genes involved in dissimilatory nitrate reduction (DNR) were enriched in the small pores of both prairie and switchgrass systems (Supplementary Fig. S7b), while denitrification-related genes were enriched only in the prairie system. This result is also suggestive of greater N deficiency in the small pores of the switchgrass system as compared with prairie, because DNR occurs at higher C:N ratios than denitrification[56,57].

Small-pore environments of the switchgrass and bare soil were also deficient of readily available C, likely stimulating the microorganisms to employ a variety of frugal C-use strategies. The dominance of genes associated with the later steps in the glycolysis/gluconeogenesis pathways, e.g., dihydrolipoamide dehydrogenase as activated in the small pores of the switchgrass in the 30-day incubation (Supplementary Figs. S5b and S6b) and in bare soil in the 24-h incubation (Supplementary Figs. S5a and S6a), suggests that residents of the small pores were more likely to consume the $^{13}$C labeled products of initial utilization, supplementing Acetyl-CoA (Supplementary Fig. S5b) and Succinyl-CoA (Supplementary Fig. S6b). Enriched genes for carbon-monoxide dehydrogenase suggest that the organisms were striving to obtain extra C resources (e.g., $CO_2$ and CO), helping them to survive in an inhospitable environment[58,59].

## Conceptual model of soil microhabitats

The observed responses of bacteria taxa to glucose additions in pores of contrasting sizes enable us to (i) postulate the delineation of three distinct micro-habitat types within the soil matrix (Fig. 5) and (ii) conceptualize their key characteristics (Table 1). The micro-habitat types are:

1. large-pore (Lp) habitats, which are spatially well-connected, with prevailing oxic conditions and root-originated C sources;
2. substrate-rich small-pore (SpRich) habitats, with somewhat restricted hydraulic connectivity and oxygen availability yet with abundant supplies of C and nutrients; and
3. substrate-poor small-pore (SpPoor) habitats, with restricted connectivity, deficient in oxygen, nutrients, and C.

In our experiment, the Lp habitat is represented by the large pores containing plant roots that release rhizodeposits including easily available exudates[60,61]. High pore connectivity (Fig. 1d) provides more $O_2$ as compared to the small pores[31], and the high hydraulic connectivity (Fig. 1b) suggests greater opportunities to receive labile substrates and nutrients carried in by frequent and fast water fluxes[62,63]. Therefore, microorganisms in the large pores are accustomed to periodic inputs of labile C, including glucose in our experiment. The large pores would be even more subject to episodic glucose additions. Indeed, to protect their membranes under water shortage stress, microorganisms synthesize trehalose[64], which upon rewetting hydrolyzes into glucose[65,66]. This process is particularly relevant for large pores, since they experience greater variations in water regimes, i.e., droughts followed by quick rewetting, than do small pores[67].

The SpRich habitat is represented by the small pores of the prairie soil, vast majority of which were in the active detritusphere (Supplementary Table S2), while the SpPoor, by the small pores of switchgrass and bare soils. We surmise that the proximity to roots and POM separates the small pores into SpRich and SpPoor habitats. The SpRich habitats are, in essence, located in rhizosphere and detritusphere. When in a rhizosphere, SpRich habitats receive DOC inputs from nearby roots, mycorrhiza, and root hairs, while in the detritusphere SpRich habitats receive DOC from decomposing POM. Due to their location in relative proximity to roots and larger pores, SpRich habitats

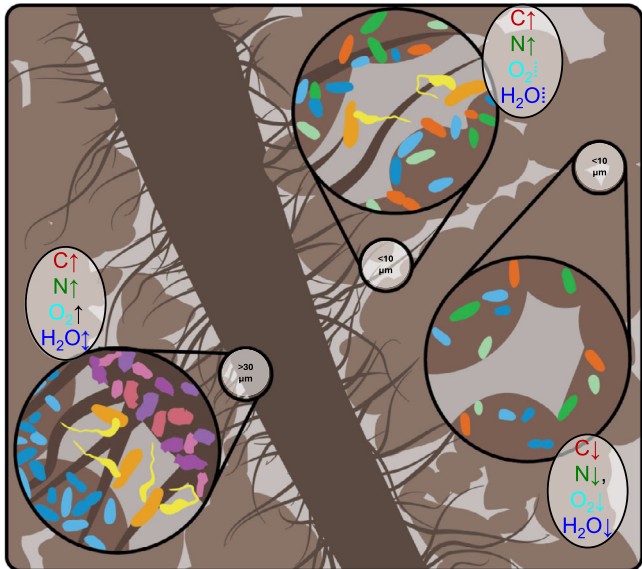

**Fig. 5 | Classification concept of soil microhabitats: large pores (Lp), substrate-rich small pores (SpRich), and substrate-poor small pores (SpPoor).** (1) The Lp habitats consist of large pores (>30 μm) that are formed and primarily occupied by roots or detritus, aka biopores, leading to high supply of C and nutrients, and O₂ availability with temporarily variable moisture conditions. The bacterial communities in Lp are dominated by plant-residue decomposers and abundant predators, and have low potential for C sequestration because of intensive decomposition of organics. (2) The SpRich habitats consist of small pores (<10 μm) in rhizosphere and detritusphere, thus with high supply of C and nutrients but somewhat limited O₂ and with a highly diverse bacterial community dominated by microbial-residue decomposers. The SpRich habitats have high potential for C sequestration because of high C input, fast microbial turnover, and close contact of microbial necromass with soil matrix. (3) The SpPoor habitats consist of small pores (<10 μm) in the bulk soil (far from roots or detritus), with limited C and nutrient resources and low O₂ leading to the bacterial community of oligotrophic passive consumers. The SpPoor habitats have low potential for C sequestration because of low C input and slow microbial turnover. Detailed description of the hypothesized characteristics of the three habitats are provided in Table 1.

also get occasional O₂ influxes. The SpPoor habitats are located in the bulk soil, getting only occasional new C and nutrient inputs with inflowing soil solution, which promotes frugal C and N use strategies by bacterial inhabitants.

We postulate that the three microhabitats differ in ecological C-acquisition strategies of their resident microorganisms. The four proposed C-acquisition groups[6] are (i) plant polymer decomposers (1° decomposers), (ii) microbial necromass decomposers (2° decomposers), (iii) predators that consume living microorganisms (predatory microbes), and (iv) passive consumers assimilating DOC, representing a spectrum of life strategies from oligotrophic to copiotrophic[6]. Our findings suggest that:

(1) Plant polymer decomposers outnumber microbial necromass decomposers in the Lp habitats, while the converse is true for SpRich habitats. Several of the taxa found exclusively in the large pores (Supplementary Table S7), i.e, *Celvibrio*, *Chitinophaga*, and *Sphingobium*, are known to be capable of plant-residue consumption via cellulolysis, lignolysis, and chitinolysis (FAPROTAX[68]).

(2) Predators are more abundant in Lp than in SpRich habitats and are very rare or absent in SpPoor[69], (refs in ref. [67]). *Bacteriovorax* was ¹³C-enriched exclusively in the large pores; and multiple members of *Bdellovibrio* taxon were among top-enriched organisms in the large pores, with only one low enriched *Bdellovibrio* OTU found in the small pores of the prairie system (Supplementary Table S6).

(3) While passive DOC consumers are present in all soil habitats, in Lp and SpRich they are fast-growing copiotrophs quickly responding to frequent inputs of labile substrates, whereas slow-growing oligotrophic microbes[70,71], adapted to low inputs, dominate the SpPoor habitat. For example, *Pseudomonas*, *Burkholderiaceae*, and *Pseudarthrobacter* are putative copiotrophs and *r*-strategists[71,72]. These organisms strongly and instantly responded to glucose addition to both large and small pores (Fig. 3b and Supplementary Table S5). Even 30 days later they remained among the abundant enriched groups in the large pores of either both or at least one of the plant systems.

Micro-environmental differences and specifics in C-acquisition strategies imply distinctive functional roles for the three microhabitats in soil C processing and protection (Table 1). Lp and SpRich habitats are locations where microbial communities are ready for active processing of frequent new C inputs. SpRich and SpPoor habitats with their large contact areas between organic compounds and mineral surfaces provide C protection. However, C protection in pores of the SpPoor habitat can be facilitated even further due to the SpPoor habitat's lower C saturation and the relative inability of its microbial inhabitants to profit from the ¹³C consumed by the so-called first-responder community[73]—the phenomenon reflected in this study's particularly high ¹³C in the small pore of the switchgrass soil (Fig. 2b). Yet, in natural conditions the inauspicious pore architecture of SpPoor habitats and their low pore connectivity limits delivery of new C. Thus, the high C protection potential of SpPoor habitats does not translate into tangible soil C gains.

The proposed *micro-habitat concept* described here is a first step towards a generalizable C processing classification of conditions within an intact, undisturbed soil matrix. While we recognize that process-based modeling of soil C cycling requires capturing the contributions of key, highly variable microbial drivers[5,65,74], thus far only a crude rhizosphere vs. bulk soil classification of such conditions has been used in modeling[75]. Our work provides experimental evidence linking physical and biochemical properties of the soil matrix with microbial functional traits and C transformations in soil microhabitats in situ. Moreover, the specific locations and sizes of the microhabitats can be quantified using X-ray μCT-based POM and pore data, thus enabling quantification and modeling of the unique physical, biochemical, biological, and ecological contributions of each habitat towards key soil functions.

Future work is needed to further test this concept. In particular there is a need to quantify microhabitats across a wide range of soil types with various textural and mineralogical characteristics, expanding to soils under specific plant communities and management practices. That said, results here suggest that this model should be fairly robust, and could represent a powerful additional way to characterize soil microhabitats in a way that also explains the distribution of microbial taxa and their important ecosystem-level processes, leading to improved biogeochemical models and perhaps management interventions.

## Methods
### Field experiment and soil sampling
Soil samples were collected from the Cellulosic Biofuel Diversity Experiment located at the Kellogg Biological Station Long-term Ecological Research Site[76], Hickory Corners, Michigan. The studied soil is a mesic Typic Hapludalf with 52% sand, 39% silt, and 9% clay[77]. The experiment is a randomized complete block (RCB) design established in 2008. For this study, we selected three plant communities that represent a range of diversities: a multiyear fallow system where soil was kept free of vegetation during 2008-2016 and then planted to corn 2016–2018; a monoculture switchgrass (variety "Southlow") system;

**Table 1 | Hypothesized physical, biochemical, and microbial community characteristics of the identified microhabitats: large pore (Lp), substrate-rich small pores (SpRich), and substrate-poor small pores (SpPoor)**

| Micro-habitat characteristics | | Micro-habitat type | | |
|---|---|---|---|---|
| | | Large pores (30–150 μm Ø) | Substrate-rich small pores (4–10 μm Ø) | Substrate-poor small pores (4–10 μm Ø) |
| Physical properties | O$_2$ availability | - Oxic conditions prevail most of the time, with anoxic micro-sites in areas of active decomposition. | - Intermittent oxic conditions, occasionally anoxic | - Anoxic conditions prevail, occasionally oxic |
| | Water availability | - Water availability is highly spatially and temporally variable: ranges from water bodies with high spatial continuity enabling microbial transport when wet to isolated menisci of varying thickness when dry; subject to frequent wet/dry cycles. | - Water is available most of the time. | - Water is available most of the time. |
| | Spatial connectivity | - Pores are well-connected.<br>- Water bodies within the pores are well-connected. | - Pores are partially connected.<br>- Water bodies are spatially fragmented. | - Pores are poorly connected.<br>- Water bodies are spatially fragmented. |
| Biochemical properties | C and N inputs as particulate and dissolved matter | - High and frequent C inputs from plant roots, detritus, and DOC from inflowing water.<br>- High N availability from plant inputs and decomposing residues. | - Medium quantities but high frequencies of C inputs from root hairs and DOC diffusing from roots and decomposing detritus as well as brought by convective water influx.<br>- Medium N availability from plant inputs and decomposing residues. | - Low and infrequent C inputs as DOC brought by convective water influx.<br>- Low N availability. |
| Bacterial communities | Diversity | Low | High | High |
| | O$_2$ status | - Strict and facultative aerobes, few to no strict anaerobes | - Facultative aerobes, few strict anaerobes | - Facultative aerobes, strict anaerobes |
| | C-acquisition strategy | - Plant-residue decomposers are more abundant than microbial-residue decomposers.<br>- Predators are abundant.<br>- Passive consumers are fast-growing copiotrophs. | - Microbial-residue decomposers are more abundant than plant-residue decomposers.<br>- Predators are nearly absent.<br>- Passive consumers are fast-growing copiotrophs. | - Microbial-residue decomposers are present.<br>- Plant-residue decomposers and predators are absent.<br>- Passive consumers are slow-growing oligotrophs. |
| Metabolic pathways | Glycolysis/citric acid cycle | - Glutamate synthesis. | - Citric acid cycle.<br>- Fatty acid synthesis. | - Wide use of degradation products.<br>- Glycolysis.<br>- Protein maintenance. |
| | Nitrogen | - When anaerobic, denitrification dominates N processing. | - When anaerobic, both denitrification and dissimilatory nitrate reduction are involved in N processing. | - When anaerobic, dissimilatory nitrate reduction dominates N processing.<br>- N fixation is used to counteract N shortages. |
| | Methane | | | Prevalent. |
| Implications for soil C storage | | - New C inputs are quickly processed, and large quantities of the new C continue to be utilized by the microbial community.<br>- The DOC products have a high potential to be carried out of the habitat by convective transport.<br>- Some DOC products may diffuse into surrounding soil matrix, to be protected from further decomposition.<br>- C processing as precursor to storage.<br>- Organic C accrual is mainly by partly processed POM, which is stored for short- and medium-term | - New C inputs are quickly processed, and large quantities of the new C continue to be utilized by the microbial community.<br>- Large quantities of DOC can diffuse into surrounding soil matrix, to be protected from further decomposition.<br>- Organic C accrual is mainly by microbial necromass, which is stored for long periods.<br>- C storage is maximal. | - New C inputs are not as efficiently and completely utilized by the microbial community as in the other habitats.<br>- Unclaimed DOC can diffuse into surrounding soil matrix, to be protected from further decomposition.<br>- Organic C accrual is solely by microbial necromass, which is stored for very long periods.<br>- C storage potential is high, but the actual C storage is low. |

Also presented are the prevalence of microorganisms with specific carbon acquisition ecological strategies[6] within each habitat, and hypothesized contributions of each habitat to soil C cycling.

and a high plant diversity system consisting of 6 native grasses and 24 native forbs typical of Michigan restored prairie communities. We refer to these systems as bare, switchgrass, and prairie, respectively. The RCB design enables us to establish causality for effects of different systems on soil and microbial characteristics insofar as the three systems were randomly assigned to 9 ×27 m plots within replicate blocks. The entire experimental area was located on a well-drained flat terrain, in a field cultivated conventionally for >100 years previous.

Three experimental plots per switchgrass and prairie vegetation system and four plots per bare system were sampled in 2019. In each plot six intact soil cores (5 cm Ø, 2.5 cm height) were collected from the 5 to 7.5 cm depth increment using a 5.7 cm diameter soil core sampler (Soilmoisture Equipment Corp., Santa Barbara, CA, USA) fitted with an acrylic sleeve. After removing the 0–5 cm top soil layer the sampler was gradually driven into the soil to collect a minimally disturbed intact core. The intact cores collected from each plot were in close proximity (<3 cm distance) to each other. The loose (bulk) soil around the cores was also collected for subsequent analyses (~500 g per plot). To prevent drying the cores were wrapped in parafilm and aluminum foil and the bulk soil was placed into zip-lock bags, all stored at 4 °C prior to analyses. Gravimetric water content was measured using a 20 g sub-sample of the bulk soil immediately upon collection. A workflow diagram is provided in Supplementary Fig. S1a.

Bulk soil analyses included total C and N via combustion analysis[78], soil pH[79], available phosphorus and potassium[80], and cation exchange capacity[81]. Root biomass in each plot was measured using the core method[82], and bulk density[83] was obtained from the spare (6th) intact core of each plot (Supplementary Fig. S1a).

## Glucose addition experiment

The $^{12}$C- and $^{13}$C-glucose addition experiment was built on the matrix potential approach of substrate additions to the soil pores of contrasting sizes[42–44]. We used three glucose addition treatments: (1) glucose dissolved in DI water added to the 4–10 μm Ø pores of the intact soil cores (referred to as the small-pore treatment), (2) dissolved glucose added to the 30–150 μm Ø pores (referred to as the large-pore treatment), and (3) a control treatment with only water added to corresponding pore sizes. We used a set of 5, out of the 6, cores collected from each experimental field plot, with one core assigned to the control treatment without glucose application; two cores assigned to the small and the other two cores assigned to the large-pore treatments, with either $^{12}$C or $^{13}$C-glucose added to each core (Supplementary Fig. S1a).

For glucose additions five cores representing a full set of glucose treatments from each experimental plot were processed simultaneously (Supplementary Fig. S1b). We used 100 mg ml$^{-1}$ solution of either $^{12}$C glucose or $^{13}$C-glucose (99 atom %). Equal volume (0.8 ml) of the solution was applied to every core, resulting in 80 mg of glucose added per core. Thus, we achieved an application rate of roughly 50 μmole C g$^{-1}$ dry soil, consistent with reported recommendations for SIP analyses[84]. It should be noted that given high variability in pore space, stone contents, root and other organic residues contents within the intact cores of this study, adding equal amounts of glucose per soil volume, as opposed to per soil mass, was regarded as the only feasible approach that would enable consistent comparisons among the studied treatments. The volume of the solution to be added (0.8 ml) was estimated as the average smallest volume of the pores of the target group, which happens to be the small pores in the switchgrass cores (Supplementary Table S2), as observed from μCT analysis. Adding this much liquid would, in theory, fill completely all target small pores in those soil cores where the volume of such pores was the smallest, while the cores with greater volumes of small pores would have some of those target pores left without the glucose solution. Likewise, since large pores occupied a much bigger soil volume than the small pores (Supplementary Table S2), only a portion of such pores has received

the glucose solution. Thus, the differences in the actual volumes of small or large pores among the studied treatments were irrelevant for the treatment comparisons—the same volumes of pores of each size group were filled with the same glucose solution in each core. It is important to point out that the only possible effect of the discrepancies between the volume of the added glucose solution and the actual volumes of the target pores in this study would be a decrease and some dampening of the differences between the pore treatments and the systems. Thus, our results, as far as comparisons between the pore treatments and systems, represent conservative estimates of the real differences that could have been obtained, if it were possible to perfectly fill all pores of the target size ranges.

To ensure glucose delivery to the pores of specific size ranges, we used the following approach (Supplementary Table S1): first, cores were drained to 400 kPa to ensure that all pores >1 μm were emptied, then water was added to bring them to 75 kPa to fill the smallest (1–4 μm) pores with water. It should be noted that while draining the soil always results in some rearrangement of soil pore space, the relatively coarse texture and non-expanding clay mineralogy of the studied soil minimized such impact. Then either water (in the large-pore and control treatments) or glucose solution (in the small-pore treatment) was added to bring the samples to 30 kPa, filling the ~4–10 μm pores. Then water was added to all cores to 10 kPa in order to create a water buffer between the target small and large pores. Finally, either water (in the small-pore and control treatments) or glucose (in the large-pore treatment) was added to 2 kPa to fill large (~30–150 μm) pores. Glucose additions were conducted in a cold room (2 °C) to minimize microbial consumption of the added glucose until it reached the target pores, after which the cores were brought to 22 °C for incubation.

To assess the microbial community involved in the immediate consumption of the added glucose and then to explore the longer-term fate of the added C, we incubated the soil cores for either 24 h or 30 days (Supplementary Fig. S1a). Each incubation period was followed by analysis of δ$^{13}$C in the soil, dissolved organic carbon (DOC), and microbial cytosol, as well as in gas samples taken during the subsequent 30-day incubation. For the first incubation the entire cores were placed into incubation jars and kept there for 24 h at 22 °C. Then the cores were cut in half, one half was passed through a 2-mm sieve, with roots and stones removed, and frozen at −20 °C for subsequent SIP and microbial biomass C analyses, with part of the soil retained to determine soil $^{13}$C (Supplementary Fig. S1a). The other half was returned to the incubation jar and incubated for 29 more days, after which it was also sieved and frozen. Gas samples for $^{13}$CO$_2$ and total CO$_2$ analyses were taken from the jars at 1, 3, 7, and 30 days during the incubation. Each incubation jar was equipped with a small water-holding container to maximize air humidity and minimize evaporation. Soil gravimetric moisture content was measured using a sub-sample after both 24 h and 30-day incubations. The average moisture contents after the 24 h and 30-day incubations were equal to 24% and 22%, respectively.

Microbial biomass was analyzed using the fumigation-extraction method[85,86]. In brief, soil samples fumigated in ethanol-free chloroform and non-fumigated samples (5 g) were subjected to 0.5 M KCl extraction (1:5 of soil:solution ratio). Upon shaking, centrifugation, and filtering (0.45 μm membrane), the extracts were freeze-dried and subjected to the total C and δ$^{13}$C analyses. The difference between fumigated and non-fumigated samples was reported as cytosol C, while the non-fumigated samples represented dissolved organic C. Measurements of δ$^{13}$C were performed in the stable isotope facility at Michigan State University using an Isoprime Vision IRMS interfaced to a Vario Isotope Cube elemental analyzer (Elementar).

## X-ray computed microtomography (μCT) scanning

One of the six cores collected at each experimental plot was used to characterize soil pores and to quantify particulate organic matter

(POM) within the intact soil matrix using X-ray μCT. Two intact mini-cores (8 mm Ø and 1 cm height) were taken from each core (Supplementary Fig. S1a), air-dried, and scanned at the Advanced Photon Source, Argonne National Laboratory (sector 13-BM-D). The scanning energy of the monochromatic beam was 24 keV, and the scanning resolution was ~4 μm. The μCT image analyses were conducted using ImageJ/Fiji[87,88]. The noise on images was removed using a 3D median filter, followed by contrast enhancement with 0.3 saturation.

Soil POM was determined using a volumetric approach of X-ray μCT, which has demonstrated good agreement with conventional POM measurements[51,89,90]. POM fragments were identified using the machine-learning-based classification approach in ilastik 1.0[91] (Fig. 1a); then the volumes of all identified POM fragments were added and the overall POM was reported as a percent of the total soil volume. Size distributions of POM fragments were obtained using the particle analyzer tool of BoneJ[92]. The smallest size of the POM fragments was set as equal to ×10 of the image resolution, e.g., $40 \times 40 \times 40$ μm, as a conservative estimate of the size of the reliably detectable μCT image features[93,94]. The identified POM was regarded as solid material in subsequent solid-pore segmentations.

The solids and pores in the images were segmented using the Otsu method as implemented in ImageJ[88]. Pore-size distributions were obtained using the maximally inscribable spheres approach as implemented in Xlib plug-in for ImageJ[95]. The connectivity of the pore space was determined as the percent of the total soil volume that was occupied by the largest cluster of interconnected pore voxels[96]. Then we also determined the volumes occupied by clusters of small (4–10 μm Ø) and large (30–150 μm Ø) pores not connected with the largest pore cluster.

We used the average distances to pores and POM, i.e., the average distances between individual voxels of the soil solid material and the border of the nearest visible (Ø >10 μm) pore or of the nearest POM fragment, as a measure of homogeneity in pore and/or POM spatial distributions. The average distances between solid soil matrix voxels and either pores or POM were obtained using a 3D distance function.

We also determine the percent of the soil matrix volume that was within the active detritusphere, i.e., in close proximity to POM fragments. While the overall distance at which roots or decomposing plant residues can influence the surrounding soil matrix can extend to 3–4 mm[97–99], here we selected <300 μm as the distance where detritus influences and microbial responses to detritus were expected to be strongest[97,100]. While in the scanned air-dried samples there were often pockets of empty pore space around the POM fragments, we assumed that under natural conditions of a relatively wet soil the POM fragments would act as sponges[101–103] expanding their volumes and filling most of the voids they are in, while the remaining empty areas around the fragments would be occupied by water films. We build on this assumption while estimating the extent of the influence of POM on the surrounding soil, which is assumed to proceed via both direct contact with minerals and through the water films.

We employed multi-energy X-ray computed tomography (μCT) scanning to visualize the specific locations of and volumes occupied by the liquids added to pores of different sizes. Specifically, to characterize the distributions of glucose solutions when added to the cores, three intact mini-cores (8 mm Ø and 5 mm height) were subjected to multi-energy X-ray μCT scanning after adding the dopant solutions (10% KI and 10% BaCl$_2$) to the small and large soil pores[104]. Since glucose at the concentrations used in this study does not precipitate and the water-glucose solution has a similar viscosity to the KI and BaCl$_2$ solutions, we assume that the distribution of glucose in soil pores does not differ significantly from those for the dopant solutions when applied in the same volumes and at the same matrix potentials. The spatial distribution of the dopants in soil pores was quantified based on the dopant mass attenuation coefficients. Even though the time-consuming and expensive nature of the analysis limited the

number of cores could be analyzed, we were able to verify where the applied solutions were placed within the soil pore system when applied to target small and large pores, as well as to characterize the hydraulic connectivity of large and small pores (Fig. 1b and Supplementary Fig. S2). Connectivity of the pore space filled with the solutions targeting small or large pores was estimated as the size of the largest cluster 90% saturated by Iodine for small pores or as the size of the largest cluster 90% saturated by Ba for large pores. This approach was proposed and first implemented in the Scamp plug-in for ImageJ[105]. Here we used Particle Analyzer function of the BoneJ plug-in to identify clusters and determine their sizes.

## DNA extraction, ultracentrifugation, and fractioning

DNA was extracted using the DNeasy PowerSoil kit or the DNeasy PowerSoil Pro kit (Qiagen, USA) following the manufacturer's protocols. Approximately 1 g of each soil was used for DNA extraction. Carnation instant nonfat dry milk (40 mg, Nestlé, Rosslyn, VA) was added at the beginning of the extraction process to improve the DNA yield[106]. DNA concentrations from each extraction were determined using the Qubit fluorometer (Thermo Fisher, USA) with the dsDNA HS Assay kit. For ultracentrifugation, approximately 10 μg of each DNA extract was mixed with Tris-EDTA buffer (10 mM Tris, 1 mM EDTA, pH 8) and cesium chloride (CsCl) solution (1.62 M) and loaded into Quick-Seal Round-Top Polypropylene tubes (13 × 51 mm, 5 ml; Beckman Coulter, USA). Refractive index (RI) values of each solution were determined using AR200 digital refractometer (Leica Microsystems Inc., Buffalo Grove, IL) and the RI was adjusted to between (1.4069–1.4071) by adding small volumes of TE buffer or CsCl solution. The sealed tubes were ultracentrifuged at 178,000×$g$ (20 °C) for 46 h in a StepSaver 70 V6 vertical titanium rotor (8 by 5.1 mL capacity) within a Sorvall WX 80 Ultra Series centrifuge (Thermo-Scientific, Waltham, MA). Following ultracentrifugation, each tube was placed onto a fraction collection system (Beckman Coulter) to generate ~26 fractions (200 μL). The RI of each fraction was determined, and, from this, buoyant density values were calculated. CsCl in the fractions was removed using linear polyacrylamide (Thermo-Scientific, USA) and a polyethylene glycol solution (1.6 M NaCl, 30% PEG solution; Thermo-Scientific, USA). The DNA concentration in each fraction was determined using the dsDNA HS Assay kit to identify the four heaviest fractions with the minimum amount of DNA for high throughput sequencing. For each of the labeled and unlabeled glucose-amended samples, sixteen tubes were ultracentrifuged: four replicate blocks for the bare soil 24 h incubation; three replicate blocks for both the switchgrass 24 hour incubation and the high diversity prairie 24 h incubations; and three replicate blocks for both the switchgrass 30-day incubation and the high diversity prairie 30-day incubations. As both small pores and large-pore incubations were also examined, in total, 64 tubes were ultracentrifuged (2 glucose forms [$^{12}$C and $^{13}$C] × 16 treatments/blocks × 2 pore sizes).

## Miseq Illumina sequencing and Mothur analysis

Total genomic DNA extracts (before ultracentrifugation) and ultracentrifugation fractions were submitted to the Research Technology Support Facility (RTSF) at MSU for 16 S rRNA gene amplicon sequencing. For each of the 64 ultracentrifugation runs (as described above), three heavy fractions (buoyant density 1.73–1.75 g/ml) and one light fraction (~1.70 g/ml) were submitted in triplicate for sequencing. This involved amplification of the V4 region of the 16 S rRNA gene using dual indexed Illumina compatible primers 515f/806r, as previously described[107]. PCR products were batch normalized using Invitrogen SequalPrep DNA Normalization plates and the products recovered from the plates pooled. The pool was cleaned and concentrated using AmpureXP magnetic beads; then QC'd and quantified using a combination of Qubit dsDNA HS, Agilent 4200 TapeStation HS DNA1000, and Kapa Illumina Library Quantification qPCR assays. The pool was

loaded onto an Illumina MiSeq v2 standard flow cell and sequencing was performed in a 2 × 250 bp paired end format using a MiSeq v2 500 cycle reagent cartridge. Custom sequencing and index primers were added to appropriate wells of the reagent cartridge.

Base calling was performed by Illumina Real Time Analysis (RTA) v1.18.54 and RTA output demultiplexed and converted to FastQ format with Illumina Bcl2fastq v2.19.1. The amplicon sequencing data in the fastq format was analyzed by Mothur[108] using the Mothur MiSeq SOP (accessed August 2021)[107]. Briefly, the Mothur analysis involved trimming the raw sequences and quality control. The SILVA bacteria database (Release 138) for the V4 region[109] was used for the alignment. Chimeras, mitochondrial, and chloroplast lineage sequences were removed, then the sequences were classified into operational taxonomic units (OTUs) at a 0.03 cutoff. The OTUs were then grouped into taxonomic levels and the downstream analysis conducted in R (version 4.0.2)[110] with RStudio (version 1.5042)[111]. The sequencing data of the total DNA and SIP fractions were submitted to NCBI under Bioproject PRJNA801760 (accession numbers SAMN25378717 to SAMN25378796) and Bioproject PRJNA802612 (accession numbers SAMN25563888 to SAMN25564655), respectively. Sequencing data from the total DNA extracts and the ultracentrifugation fractions were analyzed separately, as described below.

### Total DNA community analysis
For the analysis of the total DNA samples, two Mothur files (shared file and taxonomy file) along with an independently created metafile were used as the input for packages phyloseq[112] (version 1.34.0), file2meco (version 0.1.0)[113], and microeco (version 0.5.1)[113]. A total of 28.1 Gbytes were sequenced, with an average of 31.5 Mbytes for each sample. This resulted in the creation of (1) a phylum-level bar chart, (2) a Venn diagram of OTU abundance, and (3) two boxplots at the genus and order level. The packages phyloseq[112] (version 1.34.0), microbiome[114] (version 1.12.0) and ampvis2 (version 2.7.11)[115] were used to (1) generate heatmaps of the most abundant genera, (2) perform alpha diversity analysis (Chao1, ACE, Shannon's values, Simpson, Inverse Simpson, and Fisher indices), and (3) create barplots for the most abundant classes. The data for the alpha diversity measurements were rarefied using the phyloseq function rarefy_even_depth(pseq, sample.size = 50, rngseed = 1). The "adonis" function in the package vegan (version 2.5.7)[116] was used to test differences between microbial communities in different soil treatments with Permutational Multivariate Analysis of Variance (PERMANOVA). The "pairwise.adonis" function in package pairwiseAdonis (version 0.4)[117] was used for the comparison of significant PERMANOVA results ($P < 0.05$). The "simper" function in the package vegan (version 2.5.7)[116] was used for dissimilarity analyses for the significant comparison results ($P < 0.05$). The abundance and the classification of top twenty OTUs contributing to the difference between treatments and the connection between the OTUs and different samples was determined using the circlize package (version 0.4.13)[118].

### Identification of enriched phylotypes
Sequencing datasets were compared between the heavy and light fractions of the $^{13}$C-glucose-amended samples and fractions of similar buoyant density from the $^{12}$C glucose-amended samples to determine which phylotypes were responsible for label uptake. For this, data generated from the packages phyloseq[112] (version 1.34.0) and microbiome[114] (version 1.12.0) were analyzed using the packages dplyr (version 1.0.7)[119], tidyr (version 1.1.4)[120], ggpubr (version 0.4.0)[121] and rstatix (version 0.7.0)[122]. Specifically, those enriched in the heavy fractions of the $^{13}$C-glucose-amended samples (compared to the same fractions in the $^{12}$C glucose-amended samples) were determined using the Wilcoxon test (function wilcox_test) in RStudio (one-sided, $P < 0.05$). From those significantly enriched, the six most abundant

were selected for the creation of boxplots using ggplot2 (version 3.3.5)[123]. The analysis also included the comparison of phylotypes in the light fractions of the $^{13}$C-glucose-amended fractions compared to the light fractions of the $^{12}$C glucose-amended samples. Those enriched in the light fractions of the $^{13}$C-glucose-amended samples were removed from the above analysis to limit the possibility of reporting false positives.

### Function prediction by PICRUSt2
PICRUSt2[124] was used to predict the microbial functions of the sequencing data from the Kyoto Encyclopedia of Genes and Genomes (KEGG) orthologs (KO)[125]. Biom and fasta files generated by Mothur were used for this analysis. The PICRUSt2 analysis included sequence placement with EPA-NG[126] and gappa[127], hidden state prediction with castor R package[128] and pathway abundance inference with MinPath[129].

The KO functions associated with carbohydrate metabolism and energy metabolism were examined, including citrate cycle (TCA cycle) [PATHko00020], glycolysis gluconeogenesis [PATHko00010], methane metabolism [PATHko00680] and nitrogen metabolism [PATHko00910]. The metagenome output files were analyzed with ggplot2 (version 3.3.5)[123] and ggpubr (version 0.4.0)[121] in RStudio (version 1.5042)[111]. The relative abundance of genes associated with carbohydrate metabolism and energy metabolism were also determined for each treatment. The enriched genes in the samples amended with $^{13}$C-glucose were determined using a similar approach as described above for the enriched phylotypes (Wilcoxon test, $p < 0.05$). The eight most abundant genes associated with carbohydrate metabolism and energy metabolism were displayed in barplots. The most abundant phylotypes associated with methane metabolism and nitrogen metabolism for the total DNA samples were determined in RStudio (version 1.5042)[111].

### Statistical analysis for soil and CO$_2$ emission data
Statistical analyses were conducted using the mixed model approach implemented in *proc mixed* procedure of SAS[130]. The statistical models for all variables included the fixed effect of the plant system and pore treatment, with their interaction, and the random effect of the field replication blocks. The normality of residuals and equal variance assumptions were checked by examining normal probability plots and side-by-side boxplots, respectively. When normality assumption was found to be violated the data were log-transformed. When the equal variance assumption was found to be violated an unequal variance analysis was conducted using *repeated* option of *proc mixed*. When the main effects were statistically significant ($P < 0.05$) mean separations among the systems were conducted using *t* tests, while statistically significant interactions were followed by slicing. Differences significant at $P < 0.1$ are reported as trends. Detailed descriptions of the specific statistical models used for the studied variables along with their ANOVA results are provided in respective supplementary tables (Supplementary Tables S3 and S4).

The statistical models for the CO$_2$ emission data and other data from the incubation experiments included the incubation time and its interactions with plant systems and pore treatments, as fixed effects. The models for these data and for the data from µCT image analyses also included the random effects of experimental plots (nested within the plant systems). Time was treated as a repeated measures factor in the analyses for the $^{13}$CO$_2$ and CO$_2$ data, and appropriated repeated measures structures were selected as per Milliken and Johnson[130] (via Bayesian Information Criterion) and used for subsequent plant system and pore treatment comparisons.

### Reporting summary
Further information on research design is available in the Nature Portfolio Reporting Summary linked to this article.

## Data availability

The sequencing data of the total DNA and SIP fractions were submitted to NCBI under Bioproject PRJNA801760 (accession numbers SAMN25378717 to SAMN25378796) and Bioproject PRJNA802612 (accession numbers SAMN25563888 to SAMN25564655), respectively. The sequences were aligned using SILVA bacteria database (Release 138) obtained from the Miseq SOP (accessed August 2021). All the data is publicly released. Source data are provided with this paper.

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

## Acknowledgements

The authors thank Maxwell Oerther and Nguyen Thi Thuy Linh for invaluable assistance with laboratory experiments and soil measurements, and Mark Rivers, APS Argonne National Lab for assistance with X-ray μCT scanning. The authors are indebted to Chelsea Mamott of the Great Lakes Bioenergy Research Center communication team for the artwork. The authors thank Jonathan Alektiar for consultation on metabolic cycle results. This research used resources of the Advanced Photon Source, a U.S. Department of Energy (DOE) Office of Science user facility, operated for the DOE Office of Science by Argonne National Laboratory under Contract No. DE-AC02-06CH11357. Extraordinary facility operations were supported in part by the DOE Office of Science through the National Virtual Biotechnology Laboratory, a consortium of DOE national laboratories focused on the response to COVID-19, with funding provided by the Coronavirus CARES Act. This research was funded in part by the Great Lakes Bioenergy Research Center, U.S. Department of Energy, Office of Science, Office of Biological and Environmental Research (Award Number DE-SC0018409; A.N.K., A.K.G., and G.P.R.); by the U.S. Department of Agriculture NIFA program (grant number 2019-67019-29361; A.N.K.); by the NSF DEB Program (Award # 1904267; A.N.K.), by the NSF LTER Program (DEB 1027253) at the Kellogg Biological Station, and by Michigan State University AgBioResearch. The contribution of E.B. was motivated and made within the context of the Priority Program SPP2322 Soil Systems funded by the Deutsche Forschungsgemeinschaft (DFG, project 465122443).

## Author contributions

Z.L. conducted SIP experiment and analyses of biological data; A.N.K., A.K.G., and A.C. developed research concepts with inputs from G.P.R., Y.K., and E.B.; A.K.G. conducted the glucose addition experiment and X-ray CT scanning; A.N.K. wrote the manuscript with inputs from E.B., Y.K., and G.P.R. All authors contributed to manuscript writing and reviewed the manuscript.

## Competing interests

The authors declare no competing interests.
