## [Peer Review File · Nature Communications]

Composition and metabolism of microbial communities in soil poresREVIEWER COMMENTS

Reviewer #1 (Remarks to the Author):

The manuscript highlights the importance of soil pore-scale microbial assembly and functioning to understand the element/energy flow in soils and ecosystems. Authors used a number of contemporary techniques to demonstrate the pore-scale heterogeneity in particulate organic matter (POM), hydrolytic connectivity, and bacterial composition and function. Based on multi-lines of evidence in pore-scale heterogeneity of POM, bacteria, functional genes predicted by PICRUSt, and C 13-labeled CO₂ release, authors proposed the new classification of pores that might help incorporate soil structure into biogeochemical models and also make informed decisions of agriculture management. Overall, the study helps advance fundamental knowledge of soil structure in dictating organic matter decomposition, soil carbon sequestration and nitrogen retention. My comments and suggestions are mainly about context clarification and data interpretations.

Context clarification

1. L106: Please make a comment on whether soil perturbation by 400 kPa suction affected soil pore size distribution or bulk density. Did you measure soil bulk density or use X-ray μ Ct to examine pore size distribution following exposure of soil cores to a high pressure. Because data interpretations of “disturbed” soil cores were based on pore size distribution metrics/indices gauged on “undisturbed” soil cores, it is needed to validate no change or minimal alteration of pore size distributions. What is the soil texture?
2. L143: how did you precisely know the mass/weight of dry soil in intact cores? Suggest to use “roughly 50 μ mol C/g dry soil”. What was the range of total glucose C added into soil cores across three systems? What is the atom percent of C-13 labeled glucose? Given that glucose was added into target pores and % of target pores differed significantly among systems, I wonder if soil mass-based glucose addition would cause significant differences in glucose-C within target pores. For example, switchgrass had a greater proportion of large pores (30-150 μ m). Does that mean that the target pore volume-based concentration of glucose C was much lower in switchgrass than in prairie? If so, how such a difference in C availability could affect comparisons of microbial proliferation and activity between three systems?

Data interpretations

3. Bacterial physiological characteristics and nutrition status

L414-418: Suggest to delete “rhizosphere bacteria”. Gemmatimonadetes are present in both bulk soil and rhizosphere. To my knowledge, I am not aware that Gemmatimonadetes are always dominant in rhizosphere. Similarly, suggest to delete “plant-originated metabolites”. Many bacteria are able to decompose primary metabolites, e.g., sugars and amino acids. What do you mean plant-originated metabolites? Secondary metabolites? Given that the relative abundance of phylum Acidobacteria was similar among three systems, yet its order Acidobacteriales and genus Acidothermus were higher in bare than in vegetation soils, perhaps other factors, such as pH are more important cause. The pH value was 0.5 – 0.9 unit lower in bare than in switchgrass and prairie.

4. Soil structure effects on microbial diversity

L550-556: Besides possible differences in quantity and quality of substrates among three systems, can you discuss how differences in pore size distribution affect microbial diversity. Can physical separation minimize species competition and therefore promote diversity?

5. Size vs. composition in affecting the rate of soil CO₂ emission

L556-563: The lower CO₂ emission in switchgrass could also be caused by lower population size of the microbial community. Did you have data of microbial population or biomass before glucose addition. Glucokinase EC2.7.1.2 were not enriched in large pores of switchgrass after 24 h and 30 day incubation, but were for prairie, However, soil CO₂ emissions only differ between the two systems during the first 24 hours, but not around day 30. So it was not convincing to selectively use differences in EC2.7.1.2 between the two systems to interpret differences in CO₂ emission during 24 h. Yet completely ignore the differences in EC2.7.1.2 but similar rate of CO₂ emission after 30 days. Further, Based on the data of 24 h and 30 days, I doubt the glucokinase EC 2.7.1.2 could be enriched during day 3-7 when CO₂ emissions were much greater in switchgrass than in prairie.

6. Oligotrophs vs. copiotrophs for carbon used

L602: Did you measure soil NH₄ and NO₃ concentrations from the field plots. There is also another possibility that microbes in switchgrass and prairie differed in their carbon used efficiency: more biomass production versus more energy production – oligotrophs vs copiotrophs

Other comments

L116: What is the number of blocks? It seems three blocks as suggested by L118, am I right?

L912, 941: The reference was repeated twice; delete one.

L1001: reference is incorrect; also lacking the name of the journal

L243: Four blocks for bare? Based on Fig. S1, it seems only three blocks for bare, switchgrass and prairie.

L287: What is the sequencing depth for downstream data analysis?

L302, Delete “,”

Fig 2d: Suggest to replace Y label: cytosol with microbial biomass

L310-311: Don't understand. You did compare phylotypes in the light fraction of C-13 versus C-12 glucose amended samples. Why stating that phylotypes enriched in the light fraction of C-13 glucose amended samples were removed?

L344: Should be Table S3-S4

L395-410: Suggest to swap this paragraph with section 3.2 so that you are able to describe figures sequentially. Or you may consider to put this paragraph into the supplementary file since data were only for validation of glucose addition to target pores.

L416: were

Table S4: The second column should be prairie and switchgrass at the end of incubation (i.e., 30 days) since 24 h has been reported in the first column

L423: Based on Table S4: microbial diversity was detected to be higher in small than in large pores only at the end of incubation, but not for 24 h incubation.

L428: Is the relative abundance or C-13 enrichment?

Fig. S5: Need to label the figure panel a and b as stated in figure legend

Fig. S6 Not reported in the results, but only mention twice in the discussion. Suggest to swap with Fig. S7 or delete from the supplementary file

Fig. 3: Suggest to report the sum of relative abundance of the top 10 C-13 enriched bacterial taxa

Fig. S3-optional: not mentioned at all in the main text of the manuscript. Suggest to delete. In addition, need to indicate which one is small pores versus large pores

L529: delete (e.g., 97,

L570: any possibility that relic C13-DNA, meaning DNA was not decomposed and remain in the soil

Reviewer #3 (Remarks to the Author):

The present manuscript reports on a study combining scanning tomography of soil pore structure and organic matter with isotope-assisted microbial community analyses, which is an innovative and novel approach. Especially microbial habitats and specific functions of the microbial community within soil pores are still mostly unknown. Here the study provides new knowledge in showing that microbial communities in pores of soils from different ecosystems but also pores of large and small size within the same soil differ in community composition and in metabolic functions. I am not an expert to comment on the microbial analyses in detail, but with regard to the incubation the methods section does not provide information if and how the water regime in the cores was controlled during the 30 days of incubation. It thus remains unclear to what extent the changes in community composition and metabolic pathways over time are related to drying of the (preferentially larger) pores during the incubation as

opposed to near saturated conditions at the beginning. This should be considered in the interpretation of how added carbon was re-utilized in the decomposition of microbial necromass in large vs small pores. Also, despite the novel methodological approach, at present the manuscript rather appears as two separate studies. The experimental link between the CT results and microbial analyses is rather weak and is only presented as a conceptual overlap at the final section of the discussion and in Table 1. Considering that this study is the first of its kind, with a rather limited set of samples, the strongly generalized concept presented here is not fully supported by the experimental data. In fact, as the data on microbial communities were obtained for specific pore size classes, I would suggest that the proximity of POM and respective pore space (or pore wall surface) should be more relevant for the interpretation of the data than the proximity of POM and total soil volume.

Additional comments:

L117: This implies that also the prairie soil was only taken out of cultivation 10 years prior to the sampling? If correct, this should be indicated throughout the manuscript, e.g. as restored prairie or similar term, to avoid misinterpretation.

L157: how was DOC and microbial carbon extracted?

L185-190: What was the size range used to assign a particle as POM? Did you quantify the overlap of POM and pores (e.g. root material still in a pore)? And how does POM in pores affect the calculation of soil volume in proximity of POM and vice versa?

L332: Statistical analyses for microbial data have been described in the sections before. Which data do these analyses here refer to?

L412 and following sections: why are there no OTU and phylotype data for the bare soil after 30 days?

L514: spread?

L555: But higher microbial diversity apparently only occurred in the large pores of the prairie soil, the small pores show rather similar diversity levels across all soils.

L565 and following: This appears contradictory to me. In the previous section you suggested a fast depletion in oxygen resulted in dominance of anaerobic organisms during the first hours of incubation. But in this section, you describe growth of aerobic organisms as a continued trend, whereas anaerobic growth is suggested as indicative of a shift in environmental conditions and community composition since the beginning of the incubation.

L637 and following: As you distinguish substrate rich and substrate poor pores along the lines of prairie vs switchgrass or bare soil, you may also need to consider other factors, such as management intensity or different microbial communities induced per se by the different plant communities and management levels. These will contribute to the differences observed in the analyses of community and metabolic function done here. A generalization based on the proximity of POM and soil volume obtained solely from the analyses of separate ecosystems may thus not be as robust as you suggested.

Figure 4 is not mentioned in the text.

Table 1:

- Oxygen supply: if pore diameter is the only criterion, why would there be intermittent oxic conditions in SpRich but not in SpPoor? Where is the oxygen coming from? Shouldn't it rather be that due to higher microbial activity in substrate rich small pores, oxygen is depleted faster, hence less frequent oxic conditions than in substrate poor pores with lower levels of microbial activity?

- Water availability/spatial variability in Lp: highly spatially variable water availability and high spatial connectivity of water bodies are contradictory arguments
- Chemical properties: your data do not suggest strong variation in C:N ratio of the soil, the C:N ratio of the substrate (plant residues, etc) were not determined, but can be assumed to not vary strongly either given the small variation in soil C:N ratio
- Methane: not discussed in the text at all, why only in SpPoor?
- C storage: from your study you do not have any evidence on redistribution of C from the pore into the matrix soil

Response to review comments*

*Our responses (regular font) to all reviewers' comments (**bold**) on a point-by-point basis.

Reviewer #1 (Remarks to the Author):

Reviewer: The manuscript highlights the importance of soil pore-scale microbial assembly and functioning to understand the element/energy flow in soils and ecosystems. Authors used a number of contemporary techniques to demonstrate the pore-scale heterogeneity in particulate organic matter (POM), hydrolytic connectivity, and bacterial composition and function. Based on multi-lines of evidence in pore-scale heterogeneity of POM, bacteria, functional genes predicted by PICRUSt, and C 13-labeled CO₂ release, authors proposed the new classification of pores that might help incorporate soil structure into biogeochemical models and also make informed decisions of agriculture management. Overall, the study helps advance fundamental knowledge of soil structure in dictating organic matter decomposition, soil carbon sequestration and nitrogen retention. My comments and suggestions are mainly about context clarification and data interpretations.

Response: We would like to thank the reviewer for the positive assessment of the study.

Reviewer: Context clarification

Reviewer: 1. L106: Please make a comment on whether soil perturbation by 400 kPa suction affected soil pore size distribution or bulk density. Did you measure soil bulk density or use X-ray μ Ct to examine pore size distribution following exposure of soil cores to a high pressure. Because data interpretations of “disturbed” soil cores were based on pore size distribution metrics/indices gauged on “undisturbed” soil cores, it is needed to validate no change or minimal alteration of pore size distributions. What is the soil texture?

Response: We agree with the Reviewer that any changes in soil water content (either draining or adding water) may lead to fluctuations in soil pore space structure and bulk density. We would like to point out that, first, the studied soil is relatively coarse textured and also is low on expanding clays, thus such fluctuations in this soil are smaller than what can be expected in many other soils, e.g., fine textured, smectite clay-based ones. Second, we would like to emphasize that the water-driven modification of the soil pore space is a natural process constantly on-going in undisturbed soils in the field. Thus, we see our experimental steps of water draining/addition operating within the boundaries of this natural phenomenon.

Nevertheless, per Reviewer's request, we added a comment to this effect in the revised manuscript: “*It should be noted that while draining the soil always results in some rearrangement of soil pore space, the relatively coarse texture and non-expanding clay mineralogy of the studied soil minimized such impact.*” (ll. 525-527 in the revised manuscript).

The bulk density reported in the study was measured using a separate set of cores, not the ones involved in the glucose-addition experiment. We added the clarification to the revised manuscript: “*bulk density (Grossman and Reinsch, 2002) was obtained from the spare (6th) intact core of each plot (Fig. S1a).*” (ll. 484-485 in the revised manuscript).

We added the information about soil texture: “*The studied soil is a mesic Typic Hapludalf with 52% sand, 39% silt, and 9% clay (Lee et al., 2023)*” (l. 460 in the revised manuscript).

Reviewer: 2. L143: how did you precisely know the mass/weight of dry soil in intact cores?

Response: The weights of the soil within the intact cores were determined from the core weights at the time of sampling, the ring holder weights, and while accounting for the gravimetric water content at the time of sampling. The stone-free weight of the soil within each core was determined only after the destructive sampling and stone and root/residue removal post incubation.

As we clarify in the revised manuscript, equal amount of glucose was added to each soil core: *“For glucose additions five cores representing a full set of glucose treatments from each experimental plot were processed simultaneously (Fig. S1b). We used 100 mg ml⁻¹ solution of either ¹²C glucose or ¹³C glucose (99 atom %). Equal volume (0.8 ml) of the solution was applied to every core, resulting in 80 mg of glucose added per core. Thus, we achieved an application rate of roughly 50 μmol C g⁻¹ dry soil, consistent with reported recommendations for SIP analyses (Chen and Murrell, 2010). It should be noted that given high variability in pore space, stone contents, root and other organic residues contents within the intact cores of this study, adding equal amounts of glucose per soil volume, as opposed to per soil mass, was regarded as the only feasible approach that would enable consistent comparisons among the studied treatments.”* (ll. 497-505 in the revised manuscript).

We also added the information about the gravimetric water content analyses: *“Gravimetric water content was measured using a 20 g sub-sample of the bulk soil immediately upon collection”* (ll. 480-481 in the revised manuscript).

Reviewer: Suggest to use “roughly 50 μmol C/g dry soil”.

Response: As per Reviewer’s suggestion we added “roughly” to the glucose addition rate (l. 501 in the revised manuscript).

Reviewer: What was the range of total glucose C added into soil cores across three systems?

Response: Each core received 80 mg of either ¹²C or ¹³C glucose (l. 500 in the revised manuscript).

Reviewer:

Response: The 99 atom % ¹³C glucose was used (line 499 in the revised manuscript).

Reviewer: Given that glucose was added into target pores and % of target pores differed significantly among systems, I wonder if soil mass-based glucose addition would cause significant differences in glucose-C within target pores. For example, switchgrass had a greater proportion of large pores (30-150 μm). Does that mean that the target pore volume-based concentration of glucose C was much lower in switchgrass than in prairie? If so, how such a difference in C availability could affect comparisons of microbial proliferation and activity between three systems?

Response: We apologize for the confusion in our original description of the glucose addition process. As per above response, equal quantities of glucose solution were added to all cores. The volume of the solution (0.8 ml) was estimated as the smallest volume of the small pores within a core observed from μCT analysis (Table S2). Adding this much liquid would, in theory, fill completely all target small pores in those soil cores where the volume of such pores was the smallest, while the cores with greater volumes of small pores would have some of those target pores left without the glucose solution. Likewise, since

large pores occupied a much bigger soil volume than the small pores (Table S2), only a portion of such pores has received the glucose solution. Thus, the differences in the actual volumes of small or large pores among the studied treatments were irrelevant for the treatment comparisons – the same volumes of pores of each size group were filled with the same glucose solution in each core.

It is important to point out that the only possible effect of the discrepancies between the volume of the added glucose solution and the actual volumes of the target pores in this study would be a decrease and some dampening of the differences between the pore treatments and the systems. Thus, our results, as far as comparisons between the pore treatments and systems, represent conservative estimates of the real differences that could have been obtained, if it were possible to perfectly fill all pores of the target size ranges.

We included the description of all these considerations in the revised manuscript: ll. 505-521 in the revised manuscript.

Reviewer: Data interpretations

Reviewer: 3. Bacterial physiological characteristics and nutrition status

L414-418: Suggest to delete “rhizosphere bacteria”. Gemmatimonadetes are present in both bulk soil and rhizosphere. To my knowledge, I am not aware that Gemmatimonadetes are always dominant in rhizosphere. Similarly, suggest to delete “plant-originated metabolites”. Many bacteria are able to decompose primary metabolites, e.g., sugars and amino acids. What do you mean plant-originated metabolites? Secondary metabolites? Given that the relative abundance of phylum Acidobacteria was similar among three systems, yet its order Acidobacteriales and genus Acidothermus were higher in bare than in vegetation soils, perhaps other factors, such as pH are more important cause. The pH value was 0.5 – 0.9 unit lower in bare than in switchgrass and prairie.

Response: By “plant-originated metabolites”, we mean for example, cellulose and hemicelluloses. We surmise that organic compounds enter the soil predominantly as plant materials serving as substrates for microorganisms, which are then microbially transformed into cell constituents or excreted by cells as labile metabolic products (Bradford, 2016). Products from living microorganisms, and particularly residues of dead microorganisms (necromass), serve as a secondary source of soil organic substrates (Miltner et al., 2012; Kallenbach et al., 2016; Buckeridge et al., 2020). However, to avoid misunderstanding, we removed the term ‘plant-originated metabolites’ from the text.

We also added a mention of the potential impact of pH differences in the revised manuscript as: “*Compared to the bare soil, the relative abundance of phyla Latescibacteria, Gemmatimonadetes, Planctomycetes, Proteobacteria and Verrucomicrobia (Fig. S4) was greater in the prairie and switchgrass systems. In contrast, the relative abundances of oligotrophic (Acidobacteriales), slow-growing (Frankiales), pseudo-mycelium-forming (Actinomycetota), spore-forming (Ktedonobacterales), and desiccation-resistant (Firmicutes) bacteria were higher in bare versus planted soil. Lower pH of the bare soil might have been a contributor to a greater abundance of Acidothermus.*” (ll. 159-165 in the revised manuscript).

Reviewer: 4. Soil structure effects on microbial diversity

L550-556: Besides possible differences in quantity and quality of substrates among three systems, can you discuss how differences in pore size distribution affect microbial diversity. Can physical separation minimize species competition and therefore promote diversity?

Response: We now added the mention of pore space heterogeneity as one of the types of niche partitioning to the Discussion (l. 302-304 in the revised manuscript).

Reviewer: 5. Size vs. composition in affecting the rate of soil CO₂ emission

L556-563: The lower CO₂ emission in switchgrass could also be caused by lower population size of the microbial community. Did you have data of microbial population or biomass before glucose addition.

Response: We would like to thank the Reviewer for a valuable point. We have now added the data on soil microbial biomass C (Table S2). It is indeed higher in the restored prairie soil, which we pointed out in the revised manuscript: “*Microbial biomass was higher in the prairie soil than in the soils of bare and switchgrass systems (Table S2).*” (ll. 169-171 in the revised manuscript). We now also added the mention of the size of the microbial community as one of the explanatory factors in the Discussion (ll.296-307 in the revised manuscript).

Reviewer: Glucokinase EC2.7.1.2 were not enriched in large pores of switchgrass after 24 h and 30 day incubation, but were for prairie, However, soil CO₂ emissions only differ between the two systems during the first 24 hours, but not around day 30. So it was not convincing to selectively use differences in EC2.7.1.2 between the two systems to interpret differences in CO₂ emission during 24 h. Yet completely ignore the differences in EC2.7.1.2 but similar rate of CO₂ emission after 30 days. Further, Based on the data of 24 h and 30 days, I doubt the glucokinase EC 2.7.1.2 could be enriched during day 3-7 when CO₂ emissions were much greater in switchgrass than in prairie.

Response: We would like to respectfully clarify this point. Indeed, the differences in EC2.7.1.2 were present in both 24 hr and 30 d incubation data. However, please note that here we refer to the consumption of the glucose; and it is only very shortly after the start of the experiment (that is, for our 24 hr incubation data), the quantities of glucose available in the system were sizeable enough to make direct glucose consumption relevant to and reflected in the CO₂ emission data. Glucose is known to be consumed within hours of its addition to the soil and any glucose newly generated in a course of soil biochemical processes is very rapidly utilized (Gunina and Kuzyakov, 2015). Thus, it is highly unlikely that after 30 days of incubation there would be any sizeable quantities of the original ¹³C-labeled glucose in the system to be of any relevance to CO₂ emission results. Moreover, it was expected that the lack of available substrates in 30 days of incubation resulted in similar and low CO₂ emission in both systems regardless the employed metabolic pathways. Nevertheless, the consistency in the differences in EC2.7.1.2 at both 24 hr and 30 d incubation times gives us confidence that these results do represent an actual metabolic difference between the microbial communities of the two systems and we would like to keep the mention of that in the Discussion.

Reviewer: 6. Oligotrophs vs. copiotrophs for carbon used

L602: Did you measure soil NH₄ and NO₃ concentrations from the field plots. There is also another possibility that microbes in switchgrass and prairie differed in their carbon used efficiency: more biomass production versus more energy production – oligotrophs vs copiotrophs

Response: We understand the Reviewer’s point, but are not able to support it by experimental data because we did not measure carbon use efficiency. We assume, therefore, that discussing biomass production versus energy production influences in relation to N availability here would be too speculative. The presence of potentially oligotroph and copiotroph groups has been already mentioned in the manuscript’s conceptual model portion of the Discussion.

Unfortunately, we have no data on available soil N, which were regarded to be unfeasible to collect. However, soil N levels in these perennial systems are typically very low, as can be inferred based on studies in adjacent experiments on the same soil type (Millar and Robertson, 2015)

Reviewer: Other comments

Reviewer: L116: What is the number of blocks? It seems three blocks as suggested by L118, am I right?

Response: That is correct. The entire experiment consisted of 4 experimental blocks, but only 3 were sampled for this study. To avoid unnecessary confusion, we prefer just to specify the latter as following in the revised manuscript: *“Three experimental plots per switchgrass and prairie vegetation system and four plots per bare system were sampled in 2019.”* (ll. 471-472 in the revised manuscript).

Reviewer: L912, 941: The reference was repeated twice; delete one.

Response: Corrected.

Reviewer: L1001: reference is incorrect; also lacking the name of the journal

Response: Corrected.

Reviewer: L243: Four blocks for bare? Based on Fig. S1, it seems only three blocks for bare, switchgrass and prairie.

Response: We corrected Fig. S1 and the text as following: *“Three experimental plots per switchgrass and prairie vegetation system and four plots per bare system were sampled in 2019.”* (ll. 471-472 in the revised manuscript).

Reviewer: L287: What is the sequencing depth for downstream data analysis?

Response: The following was added to the manuscript: *“A total of 28.1 Gbytes were sequenced, with an average of 31.5 Mbytes for each sample”*. (ll. 676-677 in the revised manuscript).

Reviewer: L302, Delete “;”

Response: Corrected.

Reviewer: Fig 2d: Suggest to replace Y label: cytosol with microbial biomass

Response: Because of uncertainty in defining the conversion factor values for the studied soils, we would prefer to report the cytosol results as opposed to converting them to microbial biomass (Glanville et al., 2016; Gunina et al., 2017).

Reviewer: L310-311: Don't understand. You did compare phylotypes in the light fraction of C-13 versus C-12 glucose amended samples. Why stating that phylotypes enriched in the light fraction of C-13 glucose amended samples were removed?

Response: Yes, we also compared the light fractions of ^{13}C versus ^{12}C glucose amended samples. If phylotypes were enriched in both light and heavy fractions then these could represent potential false positives (because the enrichment could be a result of an overall large abundance of that phylotype). The sentence was changed to the following: *“Those enriched in the light fractions of*

the ^{13}C glucose amended samples were removed from the above analysis to limit the possibility of reporting false positives.” (ll. 704-706 in the revised manuscript).

Reviewer: L344: Should be Table S3-S4

Response: Corrected.

Reviewer: L395-410: Suggest to swap this paragraph with section 3.2 so that you are able to describe figures sequentially. Or you may consider to put this paragraph into the supplementary file since data were only for validation of glucose addition to target pores.

Response: Swapped, as suggested.

Reviewer: L416: were

Response: The text was edited as per earlier comment.

Reviewer: Table S4: The second column should be prairie and switchgrass at the end of incubation (i.e., 30 days) since 24 h has been reported in the first column

Response: The table is correct as is. We added the following explanation to the table’s caption to avoid confusion: “*Note that because of COVID19 lab closures bare soil samples were not subjected to the 30-day incubation. Therefore, two analyses were conducted: the first analysis was for just 24 hr data, which included all three soils and enabled comparisons between the bare soil and the other two system; the second analysis was conducted for combined 24 hr and 30 day data of switchgrass and prairie systems and enabled assessment of the time effect in these two systems.*”

Reviewer: L423: Based on Table S4: microbial diversity was detected to be higher in small than in large pores only at the end of incubation, but not for 24 h incubation.

Response: Corrected, as following: “*Across both incubation times the microbial diversity was higher in the prairie than in switchgrass soil, and at the end of the incubation it was higher in the small than in the large pores (Fig. 3a, $p < 0.05$).*” (ll. 167-169 in the revised manuscript).

Reviewer: L428: Is the relative abundance or C-13 enrichment?

Response: The sentence refers to ^{13}C enrichment (consistent to Table S5 and Fig. 3c).

Reviewer: Fig. S5: Need to label the figure panel a and b as stated in figure legend

Response: Corrected in this and other similar figures.

Reviewer: Fig. S6 Not reported in the results, but only mention twice in the discussion. Suggest to swap with Fig. S7 or delete from the supplementary file

Response: The figure contributes to the Discussion, thus we would prefer to keep it in the manuscript. We now referred to it in the revised manuscript: “*The genes encoding the enzymes involved in dihydrolipoamide dehydrogenase [EC:1.8.1.4] were activated in the small pores of in bare soil in the 24 h incubation (Fig. S5a and S6a).*” (ll. 213-215 in the revised manuscript).

Reviewer: Fig. 3: Suggest to report the sum of relative abundance of the top 10 C-13 enriched bacterial taxa

Response: We added sums of relative abundances for enriched organisms in Fig. S4d in the revised manuscript.

Reviewer: Fig. S3-optional: not mentioned at all in the main text of the manuscript. Suggest to delete. In addition, need to indicate which one is small pores versus large pores

Response: Deleted the figure as suggested.

Reviewer: L529: delete (e.g., 97,

Response: Corrected.

Reviewer: L570: any possibility that relic C13-DNA, meaning DNA was not decomposed and remain in the soil

Response: It is possible that some of ^{13}C -DNA was not completely degraded or a portion of it remained in the soil within microbial necromass. Yet, given relatively long incubation time (30 days) and known fast decomposition of labile organics, we surmise that relic intact ^{13}C -DNA would be negligible as compared to DNA remained in dormant cells or in active microbes. We hypothesize that some relic DNA could be protected from microbial decomposers in small rather than in the large pores. In contrast to this hypothesis, only two top-taxa remained the same in switchgrass at 30 days vs 24 h, suggestive of a potential occurrence of relic DNA (Fig. 3 c). In prairie, however, the only one top-taxa continued to be ^{13}C enriched at 30 days, but it actually increased in relative abundance from 24 h to 30 days suggesting active growth rather than a relic DNA storage.

Reviewer #3 (Remarks to the Author):

Reviewer: The present manuscript reports on a study combining scanning tomography of soil pore structure and organic matter with isotope-assisted microbial community analyses, which is an innovative and novel approach. Especially microbial habitats and specific functions of the microbial community within soil pores are still mostly unknown. Here the study provides new knowledge in showing that microbial communities in pores of soils from different ecosystems but also pores of large and small size within the same soil differ in community composition and in metabolic functions.

Response: We would like to thank the reviewer for the positive assessment of the study.

Reviewer: I am not an expert to comment on the microbial analyses in detail, but with regard to the incubation the methods section does not provide information if and how the water regime in the cores was controlled during the 30 days of incubation. It thus remains unclear to what extent the changes in community composition and metabolic pathways over time are related to drying of the (preferentially larger) pores during the incubation as opposed to near saturated conditions at the beginning. This should be considered in the interpretation of how added carbon was re-utilized in the decomposition of microbial necromass in large vs small pores.

Response: We have taken extra measures to minimize water losses during the incubation and monitored soil moisture; and are happy to report that the drying of the cores was so minor that its impact on the microbial communities would not be detectable in this study. We added the following clarification: *“Each incubation jar was equipped with a small water holding container to maximize air humidity and minimize evaporation. Soil gravimetric moisture content was measured using a sub-sample after both 24 h and 30-day incubations. The average moisture contents after the 24 h and 30-day incubations were equal to 24 % and 22 %, respectively.”* (ll. 545-549 in the revised manuscript).

Reviewer: Also, despite the novel methodological approach, at present the manuscript rather appears as two separate studies. The experimental link between the CT results and microbial analyses is rather weak and is only presented as a conceptual overlap at the final section of the discussion and in Table 1.

Response: As the Reviewer correctly points out: currently there is no methodology that would allow direct measurements of microbes at microscales in intact soil structure. Our study aims at joint analyses of physical and microbiological soil characteristics, attempting to view processes taking place within an intact soil body in its entirety. A few points we would like to make: 1) The measurement tools and approaches used for biological and physical measurements are completely different. 2) The tools that we used for both physical and biological analyses are novel and sophisticated. 3) By necessity, a lot of the analyses, i.e., all microbiological measurements, are destructive, thus physical and biological analyses cannot be conducted on the same sample. Thus, a comprehensive description of all the steps and an in-depth reporting of all the results are required for providing all necessary information and details on these disparate components of the intact soil system. That said, and as noted by the reviewer, we join these two paths in the conceptual model at the beginning of the paper and in the discussion where they can

be jointly interpreted to support conclusions about interactions between structural and microbial attributes. To further emphasize the connectedness of the studied concepts we now added a conceptual figure (Fig. 5 in the revised manuscript) visualizing physical and biological aspects of the soil microenvironmental space.

Reviewer: Considering that this study is the first of its kind, with a rather limited set of samples, the strongly generalized concept presented here is not fully supported by the experimental data. In fact, as the data on microbial communities were obtained for specific pore size classes, I would suggest that the proximity of POM and respective pore space (or pore wall surface) should be more relevant for the interpretation of the data than the proximity of POM and total soil volume.

Response: We agree with the Reviewer on this point, and we would like to emphasize that this is exactly why we never claimed that the proposed concept is final or broadly generalizable. In fact, we strongly emphasized the need for further in-depth testing of the proposed concept at a wide range of soils and conditions in the concluding paragraph of the manuscript, though perhaps not sufficiently strongly. We now state: *“Future work is needed to further test this concept. In particular there is a need to quantify micro-habitats across a wide range of soil types with different textural and mineralogical characteristics, expanding to soils under different plant communities and management practices. Likewise, exploring subsoil horizons will further widen the breadth of the micro-habitat characterizations.”* (ll.701-708 in the original and ll.447-451 in the revised manuscript).

Second, while we agree that it would be of some interest to explore the proximity of POM to pore walls (as the Reviewer suggests), yet we doubt that this approach would be more meaningful than the one we used. The main problem is that the empty pore space surrounding POM presents a barrier that limits both the contact of microorganisms with the POM food source and the opportunities for diffusion of POM decomposition products into the surrounding soil. The body of literature presenting experimental evidence of detritusphere extent and properties is fairly large, e.g. (Gaillard et al., 1999; Gaillard et al., 2003), and all of it points at liquid diffusion and movement of microorganisms as the drivers of detritusphere formation. Both of these drivers require direct contact between POM and soil matrix, and this is what we purposefully focused on estimating in this work. We believe, that the effect of the distances through the empty pore space, which the Reviewer suggests focusing on, would have a much lower, if any, impact.

Reviewer: Additional comments:

Reviewer: L117: This implies that also the prairie soil was only taken out of cultivation 10 years prior to the sampling? If correct, this should be indicated throughout the manuscript, e.g. as restored prairie or similar term, to avoid misinterpretation.

Response: In fact, the prairie treatment of this study has been identified as a restored prairie throughout the manuscript. In the original manuscript the first mention of this treatment (line 37) states that it is a restored prairie. The description of this treatment (line 94 in the original manuscript) specifically states that it is a restored prairie. It is later, for brevity, we refer to this treatment as just prairie. We await editorial direction as to whether this should be expanded throughout.

Reviewer: L157: how was DOC and microbial carbon extracted?

Response: We have added the following clarification: “*Microbial biomass was analyzed using the fumigation-extraction method (Vance et al., 1987; Robertson et al., 1999). In brief, soil samples fumigated in ethanol-free chloroform and non-fumigated samples (5 g) were subjected to 0.5 M KCl extraction (1:5 of soil:solution ratio). Upon shaking, centrifugation, and filtering (0.45 µm membrane), the extracts were freeze-dried and subjected to the total C and δ¹³C analyses. The difference between fumigated and non-fumigated samples was reported as cytosol C, while the non-fumigated samples represented dissolved organic C.*” (ll. 550-555 in the revised manuscript).

Reviewer: L185-190: What was the size range used to assign a particle as POM? Did you quantify the overlap of POM and pores (e.g. root material still in a pore)?

Response: We added the following clarifications to the revised manuscript: “*The smallest size of the POM fragments was set as equal to x10 of the image resolution, e.g., 40x40x40 µm, as a conservative estimate of the size of the reliably detectable µCT image features (Vogel et al., 2010; Schluter et al., 2011). The identified POM was regarded as solid material in subsequent solid-pore segmentations.*” (ll. 573-576 in the revised manuscript).

Reviewer: And how does POM in pores affect the calculation of soil volume in proximity of POM and vice versa?

Response: As noted in the original manuscript (ll. 194-195), pore voxels were not included in distance calculations. We have now further emphasized this in the revised manuscript as: “*The average distances between solid soil matrix voxels and either pores or POM were obtained using a 3D distance function.*” (ll. 587-588 in the revised manuscript).

Reviewer: L332: Statistical analyses for microbial data have been described in the sections before. Which data do these analyses here refer to?

Response: We clarified in the revised manuscript (l. 727) changing the subtitle to “*Statistical analysis for soil and CO₂ emission data*”.

Reviewer: L412 and following sections: why are there no OTU and phylotype data for the bare soil after 30 days?

Response: As described in the original manuscript (Supplemental Table4): “*Because of COVID19 lab closures bare soil samples were not subjected to the 30-day incubation.*”

Reviewer: L514: spread?

Response: Corrected.

Reviewer: L555: But higher microbial diversity apparently only occurred in the large pores of the prairie soil, the small pores show rather similar diversity levels across all soils.

Response: Respectfully, there is a misunderstanding of the figure's depiction of large vs. small pore results. We have added a more precise referral to the figure, modifying the original statement consistent with the spirit of the Reviewer's suggestion: "*maximizing microbial diversity in its small pores (Fig. 3a)*" (ll. 303-304 in the revised manuscript).

Reviewer: L565 and following: This appears contradictory to me. In the previous section you suggested a fast depletion in oxygen resulted in dominance of anaerobic organisms during the first hours of incubation. But in this section, you describe growth of aerobic organisms as a continued trend, whereas anaerobic growth is suggested as indicative of a shift in environmental conditions and community composition since the beginning of the incubation.

Response: We are not sure we completely follow this comment. Our expectation is that large pores are commonly oxygen-enriched, while small pores are frequently oxygen-depleted. Thus, in large pores a short onset of anaerobic conditions immediately after glucose addition is replaced by aerobic conditions and stimulation of aerobes. Yet, in the small pores the conditions will continue to be anaerobic, yet without any additional C resources, stimulating the organisms engaged in anaerobic processes (the subsequent paragraph). We clarified the latter point as following: "*We surmise that because of the poor contact with the atmosphere the initial oxygen depletion due to active glucose oxidation was not relieved in the small pores. While the glucose resource disappeared, the anoxic conditions still proliferated, prompting processes such as anaerobic fermentation of pyruvate to ethanol (Fig. S5b), dissimilatory nitrate reduction (Fig. S7b), and methanogenesis (Fig. 4).*" (ll. 320-324 in the revised manuscript).

Reviewer: L637 and following: As you distinguish substrate rich and substrate poor pores along the lines of prairie vs switchgrass or bare soil, you may also need to consider other factors, such as management intensity or different microbial communities induced per se by the different plant communities and management levels. These will contribute to the differences observed in the analyses of community and metabolic function done here. A generalization based on the proximity of POM and soil volume obtained solely from the analyses of separate ecosystems may thus not be as robust as you suggested.

Response: This information was provided in the original manuscript. As noted in our response to an earlier comment, in the final paragraph of the original manuscript we specifically address these issues and emphasize the need for an in-depth testing of the proposed concept at wide range of soils and conditions: "*Future work is needed to further test this concept. In particular there is a need to quantify micro-habitats across a wide range of soil types with different textural and mineralogical characteristics, expanding to soils under different plant communities and management practices. Likewise, exploring subsoil horizons will further widen the breadth of the micro-habitat characterizations. That said, results here suggest that this model should be fairly robust, and could represent a powerful additional way to characterize soil microhabitats in a way that also explains the distribution of microbial taxa and their important ecosystem-level processes, leading eventually to improved biogeochemical models and perhaps management interventions.*" (ll.701-708 in the original and ll.447-454 in the revised manuscript).

Reviewer: Figure 4 is not mentioned in the text.

Response: Respectfully, Fig. 4 was mentioned on l. 495 and l. 499 in the original manuscript (now lines 245, 249 and 324 in the revised manuscript).

Reviewer: Table 1:

Reviewer: - Oxygen supply: if pore diameter is the only criterion, why would there be intermittent oxic conditions in SpRich but not in SpPoor? Where is the oxygen coming from? Shouldn't it rather be that due to higher microbial activity in substrate rich small pores, oxygen is depleted faster, hence less frequent oxic conditions than in substrate poor pores with lower levels of microbial activity?

Response: We surmise that the reason for the difference would be relative proximity of SpRich pores to roots and large pores surrounding the roots. We added an explanation as: "*Due to their location in relative proximity to roots and larger pores, SpRich habitats also get occasional O₂ influxes.*" (ll.397-398 in the revised manuscript).

Reviewer:- Water availability/spatial variability in Lp: highly spatially variable water availability and high spatial connectivity of water bodies are contradictory arguments

Response: We would like to respectfully disagree with the Reviewer's presentation of this section of the Table 1. The complete sentence is: "*Water availability is highly spatially and temporally variable: ranges from water bodies with high spatial continuity enabling microbial transport when wet to isolated menisci of varying thickness when dry; subject to frequent wet/dry cycles.*" The hydraulic connectivity is high when the water content is high, and it can change drastically depending on the water content. We see no contradiction.

Reviewer:- Chemical properties: your data do not suggest strong variation in C:N ratio of the soil, the C:N ratio of the substrate (plant residues, etc) were not determined, but can be assumed to not vary strongly either given the small variation in soil C:N ratio

Response: Per Reviewer's suggestion, we removed mention of C:N ration from Table 1.

Reviewer: - Methane: not discussed in the text at all, why only in SpPoor?

Response: Respectfully, there was/is an entire section on methane-related results: ll. 493-499 in the original manuscript, now lines 243-249 in the revised manuscript.

Reviewer: - C storage: from your study you do not have any evidence on redistribution of C from the pore into the matrix soil

Response: As stated in the caption of Table 1, the Implications for soil C storage section of the table presents "*hypothesized contributions of each habitat to soil C cycling*". We believe it is appropriate and generally commendable to follow the findings of one's study by proposing hypotheses and research directions for future work. We also would like to point out that our hypotheses regarding C redistribution within the soil matrix are based on a very large body of literature devoted to transport of root exudates, of decomposition products in detritusphere, and of DOC transport in soil in general. Here we built on very well-known facts about these soil

phenomena, but we more specifically attribute their magnitudes to the proposed micro-habitat types.

References:

- Chen, Y., Murrell, J.C., 2010. When metagenomics meets stable-isotope probing: progress and perspectives. *Trends in Microbiology* 18, 157-163.
- Gaillard, V., Chenu, C., Recous, S., 2003. Carbon mineralisation in soil adjacent to plant residues of contrasting biochemical quality. *Soil Biology & Biochemistry* 35, 93-99.
- Gaillard, V., Chenu, C., Recous, S., Richard, G., 1999. Carbon, nitrogen and microbial gradients induced by plant residues decomposing in soil. *European Journal of Soil Science* 50, 567-578.
- Glanville, H.C., Hill, P.W., Schnepf, A., Oburger, E., Jones, D.L., 2016. Combined use of empirical data and mathematical modelling to better estimate the microbial turnover of isotopically labelled carbon substrates in soil. *Soil Biology & Biochemistry* 94, 154-168.
- Grossman, R.B., Reinsch, T.G., 2002. 2.1 Bulk Density and Linear Extensibility, *Methods of Soil Analysis*, pp. 201-228.
- Gunina, A., Dippold, M., Glaser, B., Kuzyakov, Y., 2017. Turnover of microbial groups and cell components in soil: C-13 analysis of cellular biomarkers. *Biogeosciences* 14, 271-283.
- Gunina, A., Kuzyakov, Y., 2015. Sugars in soil and sweets for microorganisms: Review of origin, content, composition and fate. *Soil Biology & Biochemistry* 90, 87-100.
- Lee, J.H., Lucas, M., Guber, A.K., Li, X., Kravchenko, A.N., 2023. Interactions among soil texture, pore structure, and labile carbon influence soil carbon gains. *Geoderma* 439, 116675.
- Millar, N., Robertson, G.P., 2015. Nitrogen transfers and transformations in row-crop ecosystems, In: Hamilton, S.K., Doll, J.E., Robertson, G.P. (Eds.), *The Ecology of Agricultural Landscapes: Long-Term Research on the Path to Sustainability*. Oxford University Press, New York, New York, USA, pp. 213-251.
- Robertson, G.P., Coleman, D.C., Bledsoe, C.S., Sollins, P., 1999. *Standard soil methods for long-term ecological research*. Oxford University Press, Oxford, UK.
- Schluter, S., Weller, U., Vogel, H.J., 2011. Soil-structure development including seasonal dynamics in a long-term fertilization experiment. *Journal of Plant Nutrition and Soil Science* 174, 395-403.
- Vance, E.D., Brookes, P.C., Jenkinson, D.S., 1987. An extraction method for measuring soil microbial biomass C. *Soil Biology and Biochemistry* 19, 703-707.
- Vogel, H.J., Weller, U., Schluter, S., 2010. Quantification of soil structure based on Minkowski functions. *Computers & Geosciences* 36, 1236-1245.

REVIEWER COMMENTS

Reviewer #1 (Remarks to the Author):

The manuscript has been considerably improved and clarified. Now I only have two minor comments

1. Reviewer: L287: What is the sequencing depth for downstream data analysis? Response: The following was added to the manuscript: "A total of 28.1 Gbytes were sequenced, with an average of 31.5 Mbytes for each sample". (ll. 676-677 in the revised manuscript).

Comment: I might not make myself clear in terms of sequencing depth. What I really meant is the rarefied sequencing depth for diversity analysis and PICRUSt2 predication.

2. Fig.3c-legend: top 10 taxa

Is it possible to specify the taxonomy-level. Are all taxa at the same taxonomy level (e.g., genus)?

Reviewer #3 (Remarks to the Author):

The authors have nicely revised the manuscript and have adressed most of the previous comments. I also apologize for overlooking the section on Methane metabolism in my previous review. However, the authors seem to have misunderstood my comments on the overlap of POM and pore space. These were in fact two separate questions:

My general comment in the beginning was directed towards the overlap of POM and microbial habitats. In your analysis you consider the overlap of POM detritosphere and soil matrix, but your microbial analyses are targeted towards communities in pores of different diameter classes. Therefore, in my opinion, it would be more in line with your interpretation (e.g. Lines 261-278 or Lines 393-400), to qunatify how much of the respective microbial habitats that you analyse by adding the 13C-label is actually within the radius of the POM detritosphere, rather than overlapping only the detritosphere with the general soil matrix for which you don't have microbial data. This would also help towards integrating the currently very separate parts of structural imaging and microbial analyses.

On the other hand, I had also asked how you consider POM in pores, for example residues of former roots still located in pores. Are these considered in the same way as POM in the soil matrix or as a separate category? Given your reply on "empty" pore space, I would very much assume the second option is needed. If POM in a pore is indeed surrounded by air-filled pore space, it should not be considered as having the same detritosphere radius as POM connected to the pore wall (or matrix soil) by water films.

In addition, two minor comments:

In Figure 1 and the respective text in the results section, it would be helpful to specify that the distances refer to the average distance from soil to pore or POM. In the current description, without reading the methods, it may be misunderstood as the distance from pore to pore or from pore to POM.

Figure 3b shows 100 OTUs for both switchgrass and bare soil instead of the 100 and 90 mentioned in the text (L175).

Response to reviewer comments:

Comments are in **bold** and responses are in regular font

REVIEWER COMMENTS

Reviewer #1 (Remarks to the Author):

The manuscript has been considerably improved and clarified. Now I only have two minor comments

Response: We are glad to hear that the reviewer is satisfied with the revised version of the manuscript.

1. Reviewer: L287: What is the sequencing depth for downstream data analysis? Response: The following was added to the manuscript: "A total of 28.1 Gbytes were sequenced, with an average of 31.5 Mbytes for each sample". (ll. 676-677 in the revised manuscript).

Comment: I might not make myself clear in terms of sequencing depth. What I really meant is the rarefied sequencing depth for diversity analysis and PICRUSt2 predication.

Response: We added the following to the revised manuscript: "*The data for the alpha diversity measurements were rarefied using the phyloseq function rarefy_even_depth(pseq, sample.size = 50, rngseed = 1).*" (ll. 688-690 in the revised manuscript).

We based our analysis on the recommendation of Gavin Douglas (the creator of PICRUSt2) on his GitHub page (under FAQ section), thus we did not rarefy sequencing data for PICRUSt2. As per above, the data for the alpha diversity measurements were rarefied using `rarefy_even_depth(pseq, sample.size = 50, rngseed = 1)`.

2. Fig.3c-legend: top 10 taxa

Is it possible to specify the taxonomy-level. Are all taxa at the same taxonomy level (e.g., genus)?

Response: Yes, the shown taxa are at genus level – we specified it now in the revised caption for Figure 3.

Reviewer #3 (Remarks to the Author):

The authors have nicely revised the manuscript and have addressed most of the previous comments. I also apologize for overlooking the section on Methane metabolism in my previous review.

Response: We are glad to hear that the reviewer is satisfied with the revised version of the manuscript.

However, the authors seem to have misunderstood my comments on the overlap of POM and pore space. These were in fact two separate questions:

My general comment in the beginning was directed towards the overlap of POM and microbial

habitats. In your analysis you consider the overlap of POM detritosphere and soil matrix, but your microbial analyses are targeted towards communities in pores of different diameter classes. Therefore, in my opinion, it would be more in line with your interpretation (e.g. Lines 261-278 or Lines 393-400), to quantify how much of the respective microbial habitats that you analyse by adding the ¹³C-label is actually within the radius of the POM detritosphere, rather than overlapping only the detritosphere with the general soil matrix for which you don't have microbial data. This would also help towards integrating the currently very separate parts of structural imaging and microbial analyses.

Response: We would like to thank the Reviewer for the insightful suggestion. We calculated the proportions of the large and small pores that were located within the “active detritosphere” and reported the results in the Supplement Table 2 of the revised manuscript.

On the other hand, I had also asked how you consider POM in pores, for example residues of former roots still located in pores. Are these considered in the same way as POM in the soil matrix or as a separate category? Given your reply on "empty" pore space, I would very much assume the second option is needed. If POM in a pore is indeed surrounded by air-filled pore space, it should not be considered as having the same detritosphere radius as POM connected to the pore wall (or matrix soil) by water films.

Response: We completely agree with the Reviewer that, ideally, the areas and extents of the direct contact between the residue fragments and the soil minerals, as well as the distances between the residues and the minerals, should be considered while determining the extent of the residue influence on detritosphere. However, we would like to point out that accounting for these influences requires unique scanning settings and can not be done using the CT data of this study. The main problem is that our scans were conducted using air-dry soil samples. Such scanning conditions are optimal for identification of soil pores and POM, however, they are not suitable to determine the contact areas between POM and soil solids. The reason is that under natural, i.e., wetter, conditions POM fragments act as sponges, absorbing water from the surrounding soil and expanding in their volumes. We have conducted a substantial amount of experimental work exploring and quantifying this phenomenon (Kravchenko et al., 2017; Kutlu et al., 2018; Kim et al., 2023). In the current study, we assume that in a relatively wet soil the expanded wet POM fragments fill most of the voids they are in, and the remaining empty areas around the fragments are occupied by water films. We added the following to clarify this assumption to the revised manuscript: “*While in the scanned air-dried samples there were often pockets of empty pore space around the POM fragments, we assumed that under natural conditions of a relatively wet soil the POM fragments would act as sponges (Kravchenko et al., 2017; Kutlu et al., 2018; Kim et al., 2023) expanding their volumes and filling most of the voids they are in, while the remaining empty areas around the fragments would be occupied by water films. We build on this assumption while estimating the extent of the influence of POM on the surrounding soil, which is assumed to proceed via both direct contact with minerals and through the water films.*” (ll. 593-599 in the revised manuscript).

While we recognize that this assumption is a simplification, nevertheless, we feel that building on this assumption in defining the active detritosphere is a more reliable approach than making post-hoc speculative decisions regarding sizes of air-gaps between dry POM and dry soil matrix

on our images and whether or not they would be present in more natural wetter soil conditions. However, again, the Reviewer brought a very important point that deserves attention as a future separate targeted research topic.

In addition, two minor comments:

In Figure 1 and the respective text in the results section, it would be helpful to specify that the distances refer to the average distance from soil to pore or POM. In the current description, without reading the methods, it may be misunderstood as the distance from pore to pore or from pore to POM.

Response: The caption for Figure 1 specifies that the distances are the average distances as following: “(c) *Average distances to pores (dark shading) and POM (light shading) in the soils of the three communities*”. We also made sure that whenever this measure is referred to in the manuscript – it is referred to as “average distance”.

Figure 3b shows 100 OTUs for both switchgrass and bare soil instead of the 100 and 90 mentioned in the text (L175).

Response: Corrected.

References

Kim, K., Kaestner, A., Lucas, M., Kravchenko, A.N., 2023. Microscale spatiotemporal patterns of water, soil organic carbon, and enzymes in plant litter detritusphere. *Geoderma* 438, 116625.

Kravchenko, A.N., Toosi, E.R., Guber, A.K., Ostrom, N.E., Yu, J., Azeem, K., Rivers, M.L., Robertson, G.P., 2017. Hotspots of soil N₂O emission enhanced through water absorption by plant residue. *Nature Geoscience* 10, 496-+.

Kutlu, T., Guber, A.K., Rivers, M.L., Kravchenko, A.N., 2018. Moisture absorption absorption by plant residue in soil. *Geoderma* 316, 47-55.

REVIEWERS' COMMENTS

Reviewer #3 (Remarks to the Author):

I have no further comments, the authors have well addressed all previously mentioned aspects.